# Improved representation of the contemporary Greenland ice sheet firn layer by IMAU-FDM v1.2G

Max Brils[1], Peter Kuipers Munneke[1], Willem Jan van de Berg[1], and Michiel van den Broeke[1]

[1]Institute for Marine and Atmospheric Research, Utrecht University, Utrecht, the Netherlands

**Correspondence:** Max Brils (m.brils@uu.nl)

**Abstract.**

The firn layer that covers 90% of the Greenland ice sheet (GrIS) plays an important role in determining the response of the ice sheet to climate change. Meltwater can percolate into the firn layer and refreeze at greater depths, thereby temporarily preventing mass loss. However, as global warming leads to increasing surface melt, more surface melt may refreeze in the firn layer, thereby reducing the capacity to buffer subsequent episodes of melt. This can lead to a tipping point in meltwater runoff. It is therefore important to study the evolution of the Greenland firn layer in the past, present and future. In this study, we present the latest version of our firn model, IMAU-FDM (Firn Densification Model) v1.2G, with an application to the GrIS. We improved the density of freshly fallen snow, the dry-snow densification rate and the firn's thermal conductivity using recently published parameterizations and by calibrating to an extended set of observations of firn density, temperature and liquid water content at the GrIS. Overall, the updated model settings lead to higher firn air content and higher 10 m firn temperatures, owing to a lower density near the surface. The effect of the new model settings on the surface elevation change is investigated through three case studies located at Summit, KAN-U and FA-13. Most notably, the updated model shows greater inter- and intra-annual variability in elevation and an increased sensitivity to climate forcing.

## 1 Introduction

Firn, the transitional stage between seasonal snow and ice in the accumulation zone of glaciers, strongly influences the climate response of mountain glaciers, ice caps and ice sheets. Pore space between snow grains that make up the firn layer enable meltwater to percolate into the firn layer, and refreeze – if the firn temperature is below freezing. This prevents runoff which means that firn acts as an efficient buffer against ice sheet mass loss. Diagnosing the current state of the Greenland ice sheet (GrIS) firn layer, and predicting its future, is therefore important in order to understand current and future changes in the mass balance of the GrIS.

A common method to assess the GrIS mass balance is altimetry. With altimetry, elevation changes are monitored by repeated scanning of the ice-sheet surface with an active laser or radar instruments on board airplanes or satellites. A crucial step in translating the observed volume change to a mass change is to determine the density of the firn associated with the elevation change. To account for variability and changes in the surface mass balance and firn processes like compaction, percolation, and refreezing, a firn model is often employed for this step (Zwally et al. (2002); Sørensen et al. (2011); McMillan et al. (2016);

Shepherd et al. (2020); Hawley et al. (2020)). This is necessary since firn densification decreases the surface elevation without changing its mass, and is affected by changes in temperature and accumulation.

Firn models can also be used to assess the evolution of the aforementioned buffer capacity of the firn layer and how it is impacted by refreezing. It has been demonstrated that refreezing is a critical process for many ice caps to survive, e.g. in the
Canadian Arctic. On these ice caps, summer melt consistently exceeds annual snowfall, and refreezing is required to maintain a near-zero mass balance (Noël et al. (2018a), Gascon et al. (2013); Bezeau et al. (2013); Ashmore et al. (2019)). As melt rates increase further in response to global warming, firn pore space is increasingly taken up by refrozen meltwater, degrading the efficiency of the refreezing process until at some point the available pore space has decreased to the extent that it cannot absorb all of the meltwater produced during summer. When this happens the rate of mass loss is increased rapidly. At Greenland's
marginal ice caps, this has already been taking place since the mid 1990s (Noël et al. (2017)). While recent research has suggested that the loss of pore space is not irreversible (Rennermalm et al. (2022)), it would generally take a much longer time to recover the lost pore space. This leads to hysteresis and it is often regarded as a tipping point.

Refreezing also plays an important role in the Greenland ice sheet (GrIS), but it has not yet reached this saturation tipping point (Pfeffer et al. (1991), Braithwaite et al. (1992)). The GrIS has an extensive firn layer ($\sim 1.4 \cdot 10^6$ km$^2$), covering about
80% of the total area of the GrIS, which is higher in elevation (on average 2100 m above sea level (a.s.l.)) and hence more porous and colder than firn on other Arctic ice caps (Noël et al. (2020)). With a depth of up to 80 m (Kuipers Munneke et al. (2015b)), Vandecrux et al. (2019) estimated that the GrIS firn layer contains a total of $26800 \pm 1840$ km$^3$ of air. This is equivalent to more than 60 times the total annual (1961-1990 average) volume of GrIS meltwater production (Van den Broeke et al. (2016)), although this is reduced to a factor of $\sim$1-4 if only pore space in the percolation zone is considered (Harper
et al. (2012)). Model estimates show that for the same period, at least 44% of the meltwater produced at the surface of the GrIS refroze in the firn layer (Van den Broeke et al. (2016); Mouginot et al. (2019)).

Surface melt is also increasing in the GrIS accumulation zone, with the extreme melt summers of 2012 and 2019 as vivid examples (Nghiem et al. (2012); Sasgen et al. (2020)). These high-melt summers led to peaks in refreezing, warming and densification of the firn layer (Steger et al. (2017a)). In some places, 1-2 m thick ice slabs were formed that prevent meltwater
from reaching the pore space below (Machguth et al. (2016); MacFerrin et al. (2019)).

Lastly, firn models can be used to interpolate between observations such as density, temperature and age of the firn (Kuipers Munneke et al. (2015b); Vandecrux et al. (2019)). This is convenient since observations from firn cores and snow pits are usually sparse and costly to obtain.

Some (regional) climate models, such as RACMO and MAR, are coupled interactively to a snow/firn module, but these
often use simplified initialization, parametrizations and/or reduced vertical resolution for computational efficiency. The main advantage of using a dedicated, offline firn densification model is the lower computational cost, which enables the use of higher vertical resolution, a proper initialization of the firn layer, and extensive sensitivity testing (Lundin et al. (2017); Stevens et al. (2020); Vandecrux et al. (2020b)). The drawback of using an offline firn model is that it must be forced unidirectionally with observed and/or modelled surface temperature and surface mass fluxes (snow, rain, sublimation, drifting snow erosion).

In this study we present an updated version of the firn densification model of the Institute for Marine and Atmospheric research Utrecht (IMAU-FDM v1.2G, henceforth IMAU-FDM) applied to the GrIS, forced at the upper boundary by the latest three-hourly output of the polar version of the Regional Atmospheric Climate Model (RACMO2, Noël et al. (2018b)). It supersedes IMAU-FDM v1.1G presented by Kuipers Munneke et al. (2015b).

    We use recently published parametrizations and previously existing and newly obtained observations of firn density, tem-
perature and liquid water content from the GrIS to calibrate model parametrizations for surface (fresh snow) density, dry snow densification rate, thermal conductivity, and meltwater percolation. The updated model is subsequently used to perform case studies of contemporary firn depth variability in three climatologically distinct locations of the GrIS accumulation zone: (1) the dry and cold interior, (2) the relatively low-accumulation western percolation zone, and (3) the high-accumulation south-eastern percolation zone.

This paper is organized as follows: in Sect. 2 we describe the model details as well as the changes made to the model. We also describe the extended set of observations, both in time and space, that allows for new parametrizations and improved calibration of IMAU-FDM for the GrIS. Then, in Sect. 3, we show how the altered model results in an overall improved representation of GrIS firn density, temperature and liquid water content. The three case studies are then presented in Sect. 4, followed in Sect. 6 by a summary and outlook. Lastly, we present an uncertainty analysis in Sect. 5.

## 2   Methods

### 2.1   IMAU-FDM

For this work we use the offline IMAU-FDM, a semi-empirical firn densification model that simulates the time evolution of firn density, temperature, liquid water content and changes in surface elevation owing to variability of firn depth. The model has been compared extensively to, and calibrated with observations of firn density and temperature from the ice sheets of Greenland
and Antarctica (Ligtenberg et al. (2011); Kuipers Munneke et al. (2015b); Ligtenberg et al. (2018)). For both the ice sheets of Greenland and Antarctica, the performance of IMAU-FDM has been comparable to the more physically-based SNOWPACK model (Steger et al. (2017b); Van Wessem et al. (2021); Keenan et al. (2021)). IMAU-FDM is forced by three-hourly output of the polar version of the Regional Atmospheric Climate Model (RACMO2.3p2) (Noël et al. (2019)). Over glaciated grid cells, the RACMO2 subsurface model uses approximately the same expressions as IMAU-FDM, but with a lower vertical resolution
(max. 150 vs. 3000 layers) and less comprehensive initialization to save computation time. The previous model version, IMAU-FDM v1.1G (Kuipers Munneke et al. (2015b)), has been forced with the same version of RACMO and has been ran at the same resolution as IMAU-FDM v1.2G. In the following subsections, we briefly describe how the main processes are currently represented in IMAU-FDM, and what improvements have been implemented compared to the previous model version.

### 2.1.1 Fresh snow density

An important boundary condition for the model is the density of freshly fallen snow, $\rho_0$. When determined from field observations, fresh snow density is often assumed equal to the near-surface density, loosely defined as the average density of the top $0.5$ m of dry snow. As density is highly variable near the surface, the exact chosen depth is critical for the outcome, which hampers a robust comparison between datasets (Fausto et al. (2018)). In firn models, fresh snow density is commonly parameterized as a function of meteorological variables such as temperature and wind speed at the time of deposition, or, when these are not available, using annual average values instead (Keenan et al. (2021)). Several studies have addressed the parametrization of $\rho_0$ on the GrIS (Kuipers Munneke et al. (2015b); Fausto et al. (2018)). Assuming a linear dependence of the density on mean annual surface temperature $T_s$, this parametrization takes on the following form:

$$\rho_0 = A + B \cdot T_s \tag{1}$$

With $A$ and $B$ being fitting constants and $T_s$ in $^\circ$C. In previous studies where IMAU-FDM was applied to the GrIS, $A = 481$ kg m$^{-3}$ and $B = 4.834$ kg m$^{-3}$ K$^{-1}$) have been used (Kuipers Munneke et al. (2015b); Ligtenberg et al. (2018)) based on observations using the top $0.2$ m average density from no-melt locations to approximate the surface value.

In the updated model, a new parameterization for fresh snow density (Fausto et al. (2018)) was adopted. In contrast to previous studies, which typically use the first $\sim 0.5 - 1.0$ m of snow, Fausto et al. (2018) used only the upper $0.1$ m of snow to define surface density at 200 locations and found:

$$\rho_0 = 362.1 + 2.78 \cdot T_a \tag{2}$$

with $T_a$ the annual mean near-surface (usually 2 m) air temperature in $^\circ$C. Previously, the climatological mean 2 m air temperature has been used in IMAU-FDM (Kuipers Munneke et al. (2015a)), or an instantaneous value (Ligtenberg et al. (2018)). In v1.2G of the model, $T_a$ is calculated as the average 2 m air temperature of the year preceding the snowfall. While the actual density of fresh snow varies on much shorter time scales than this, we opt here for a parameterization that depends on annual mean surface temperatures for three reasons.

Firstly, the parameterization is derived by fitting the measured snow densities to mean annual temperatures, not the temperature at the time of the accumulation event. Thus the equation itself links snow density to annual temperatures, not instantaneous temperatures. Therefore, using the instantaneous temperatures would introduce an additional uncertainty.

Secondly, in deriving their parameterization, Fausto et al. (2018) used the density of the upper $0.1$ m of snow. Especially in locations where only a low amount of accumulation takes place, this means that the measured layer of firn contains snow from multiple accumulation events. Moreover, it may also have compacted in the time between the accumulation event and the observation. Therefore, we believe that the typical temperature to which this $0.1$ m of snow is exposed to can more reasonably be approximated with annual temperatures than with instantaneous ones.

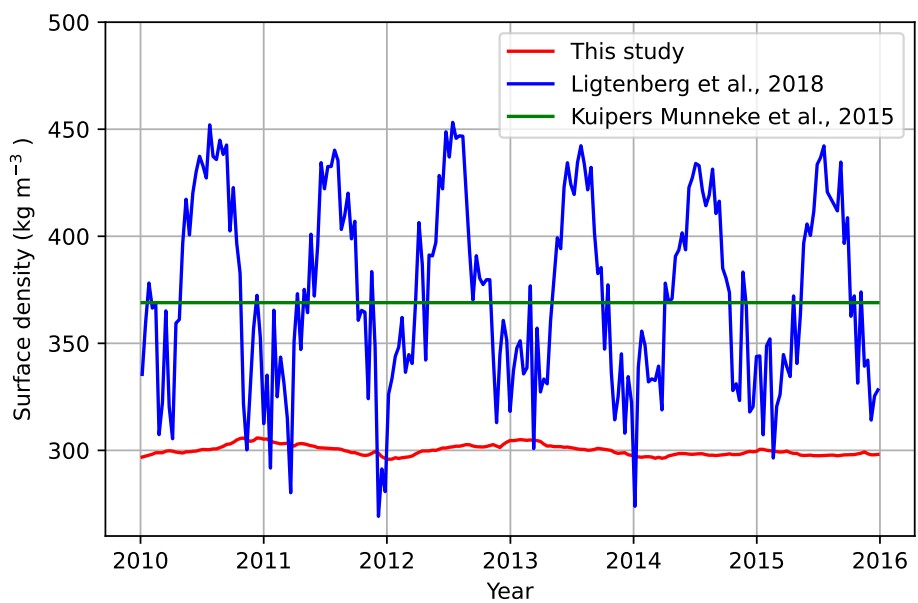

**Figure 1.** Daily averages of Das 2 (southeast Greenland, see Fig. 4) surface density (2010-2016) using three different parametrizations.

Thirdly, using a climatological mean value suppresses the year-to-year variability in snow density. This is undesirable, especially because the model will also be used future scenarios, in which long term trends in temperature may have an effect. On the other hand, using instantaneous temperature values may introduce an excessive variability which, in reality, is smoothened by the effects of the snow being subjected to settling by wind and metamorphosis through numerous, daily warming and cooling cycles. Calculating $T_a$ in the updated model as the mean annual air temperature, is a trade-off between these two extremes that allows us to account for long term trends in air temperatures.

Fausto et al. (2018) noted that surface density correlates only weakly with annual mean $T_a$ and that using a constant density of $315 \, \mathrm{kg \, m^{-3}}$ may be preferable. Therefore, we compared the model's performance while using Eq. 2, with the model's performance while using a constant density of $315 \, \mathrm{kg \, m^{-3}}$. From this we concluded that neither configuration gives significantly better results. In Sect. 3.1 this comparison will be discussed in more detail.

Fig. 1 shows the surface density (2010-2016) using three different approaches at site Das 2 in Southeast Greenland. Clearly, temporal variations are much larger when an instantaneous $T$ is used. Furthermore, the expression by Fausto et al. (2018) results in a lower surface density overall than Kuipers Munneke et al. (2015a). In subsequent sections, we refer to Ligtenberg et al. (2018)) as for the previous model version.

### 2.1.2 Dry snow densification rate

IMAU-FDM is a 1D, vertical Lagrangian model. When new snow accumulates at the surface (model top), the model layers are buried deeper and tracked during their downward motion. At every time step, each layer is compacted under the influence

of the pressure exerted by the mass of snow/firn above it. However, in IMAU-FDM the densification rate $\frac{\mathrm{d}\rho}{\mathrm{d}t}$ is not directly related to the overburden pressure, but rather follows a semi-empirical, temperature-dependent equation based on Arthern et al. (2010):

$$\frac{\mathrm{d}\rho}{\mathrm{d}t} = C\dot{b}g(\rho_i - \rho)e^{-\frac{E_c}{RT} + \frac{E_g}{RT_{ave}}} \tag{3}$$

where $\dot{b}$ is the annual average accumulation rate (mm w.e. per year) over the spinup-period (1960-1979), $\rho_i = 917$ kg m$^{-3}$ is the density of glacial ice, $g$, $E_c$, $E_g$ and $R$ are constants, and $T$ is the instantaneous layer temperature in Kelvin. Different values of $C$ above and below $\rho = 550$ kg m$^{-3}$ represent a shift in the dominant densification mechanism (Cuffey and Paterson (2011)). For $\rho < 550$ kg m$^{-3}$, the densification of the firn is dominated by the settling and sliding of grains. For $\rho \geq 550$ kg m$^{-3}$ recrystallisation, deformation and sublimation become dominant and the densification rate is lower, which is reflected

in a lower value for $C$.

     Arthern et al. (2010) base their densification rate on an equation describing Nabarro-Herring creep from Coble (1970). This rate depends linearly on the overburden pressure. The overburden pressure $\sigma(t)$ on a layer of firn of age $t_{age}$, is equal to $\sigma(t_{age}) = g \int_{t-t_{age}}^{t} \dot{b}(t\prime)dt\prime$ in the absence of melt. The accumulation rate enters the equation for the densification rate through the assumption that the accumulation rate is constant in time, replacing $\sigma(t)$. RACMO2 provides the average annual

accumulation rate $\dot{b}$ as the amount of total precipitation minus sublimation and drifting snow erosion during the spin-up period. Assuming a constant $\dot{b}$ introduces an error in the load experienced by a layer of firn ($\sigma(t_{age}) = g\dot{b}t$). However, over the time scale considered here the error remains small (e.g. $< 3.2\%$ at Summit and $< 1.9\%$ at Dye-2).

     Eq. 3 is used for both wet and dry locations, and so it is assumed that the densification rate of dry firn is equal to that of wet firn. We acknowledge that the presence of liquid water in between grains may impact the evolution of their size and shape.

This in turn may also impact the densification rate of the firn. Most firn models that account for a different densification rate of wetted firn are based of Vionnet et al. (2012). They introduce this dependency through an empirical correction factor for the firn viscosity. This correction factor is derived from experiments that have not been published (see Brun et al. (1992)). Due to a lack of physical understanding and a lack of available measurements we decided not to introduce an extra dependence of the compaction rate on the liquid water content to reduce the model's complexity and to prevent overfitting.

Compared to observations of the depth of the 550 and 830 kg m$^{-3}$ density levels, a structural bias is found when using Eq. 3, that in previous studies turned out to depend on the annual average accumulation rate. In order to calibrate Eq. 3 to the new, expanded set of observations (Sect. 2.4), we introduce a multiplication factor MO to better align modelled density profiles with observations:

$$\mathrm{MO} = \alpha - \beta \ln(\dot{b}) \tag{4}$$

where $\alpha$ and $\beta$ are unitless constants. Although the physical processes underlying the densification of firn do not explicitly depend on the accumulation rate, a correlation between $\frac{\mathrm{d}\rho}{\mathrm{d}t}$ and $\dot{b}$ may act as a proxy variable for effects that are time dependent,

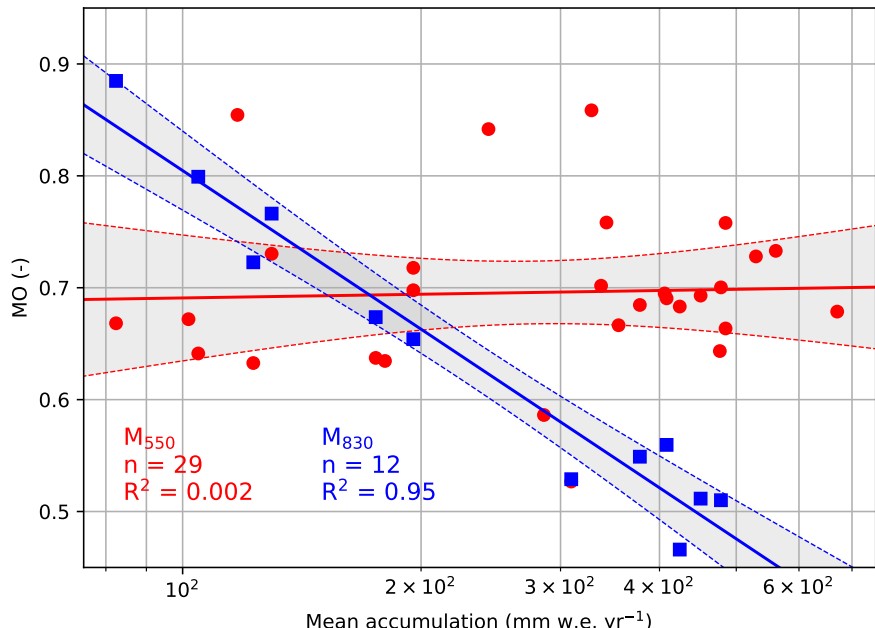

**Figure 2.** Ratio between modelled and observed depth at which the density reaches $550$ kg m$^{-3}$ (MO$_{550}$) or $830$ kg m$^{-3}$ (MO$_{830}$) as a function of local accumulation rate. The solid lines represent the corresponding regressions and the grey bands around them are their corresponding 95% confidence intervals.

possibly related to the geometry of grains (Cuffey and Paterson (2011)). Firn densification owing to horizontal compression is neglected, although in fast-flowing regions this can be locally important (Horlings et al. (2021)).

In the model update, we recalibrated the dry densification correction factor MO as a function of mean annual accumulation, by using an updated, high-resolution GrIS accumulation field (Noël et al. (2015)) and optimizing the modelled depths at which the firn density reaches the critical values $550/830$ kg m$^{-3}$ following Ligtenberg et al. (2011) and Kuipers Munneke et al. (2015b). In v1.2G the values for $\alpha$ and $\beta$ changed, but MO keeps the same general form as described in Eq. 4. To perform the new calibration, we make use of 29 cores (see Sect. 2.4). These include the 22 cores used in the previous calibration done by Kuipers Munneke et al. (2015b). Only dry firn cores (i.e. with little surface melt) are used for calibration since Eq. 3 only describes the densification rate due to overburden pressure. A core is considered as "dry" if the mean annual melt is less than 5% of the mean annual accumulation rate. At wet locations, the density at a given depth is not only impacted through compaction by overburden pressure but also through the vertical transport of liquid water inside of the firn after a melt event. This drastically alters the depth at which the firn layer reaches $550$ or $830$ kg m$^{-3}$. Using these cores for the derivation of the MO fit would incorrectly assign this density change to a different, higher, compaction rate, whereas it is due to refreezing.

Fig. 2 shows the new calibrations. The statistics of the new fit are summarised in Table 1, along with the old values from Kuipers Munneke et al. (2015b). The standard error of the coefficients is calculated by assuming that the errors in the regression

**Table 1.** Values of the old and new linear regression of Eq. 4, their $R^2$ as well as the standard error in of the new fitting parameters.

|  | $\alpha_{old}$ | $\alpha_{new}$ | $\sigma_\alpha$ | $\beta_{old}$ | $\beta_{new}$ | $\sigma_\beta$ | $R^2_{old}$ | $R^2_{new}$ |
|---|---|---|---|---|---|---|---|---|
| $MO_{550}$ | 1.042 | 0.6688 | 0.1317 | 0.0916 | -0.0048 | 0.0233 | 0.35 | 0.002 |
| $MO_{830}$ | 1.734 | 1.7465 | 0.0841 | 0.2039 | 0.2045 | 0.0154 | 0.96 | 0.946 |

are normally distributed. Least squares fitting yields $R^2$ values for $MO_{550}$ and $MO_{830}$ of $1.57 \cdot 10^{-3}$ and $0.95$ respectively. With the update and the use of new firn and accumulation data, the linear relation between $MO_{550}$ and $\ln(\dot{b})$ becomes much weaker, and the $MO_{550}$ reduces to an almost constant value of 0.67. Despite the difference with previous formulations in
IMAU-FDM, this is similar to findings by Robin (1958) and Herron and Langway (1980), who found that for densities lower than $550 \ \mathrm{kg \ m^{-3}}$ the densification rate correlates almost linearly with the accumulation rate whereas at higher densities this correlation becomes non-linear. A linear correlation between the densification rate and the accumulation rate implies a constant MO (see Eq. 3). Our correlation for $MO_{830}$ also implies that the relation between densification rate $\frac{d\rho}{dt}$ and accumulation is non-linear above $\rho = 550 \ \mathrm{kg \ m^{-3}}$. We run the model with the newly derived values for $\alpha$ and $\beta$ for $MO_{550}$ and $MO_{830}$ listed
in Table 1. By doing so, $MO_{550}$ and $MO_{830}$ retain the same general formulation.

### 2.1.3 Thermal conductivity

In IMAU-FDM, the vertical temperature distribution and its evolution is obtained by solving the one-dimensional heat transfer equation

$$\rho c \frac{\partial T}{\partial t} = -\frac{\partial G}{\partial z} + \mathcal{L} = -\frac{\partial}{\partial z}\left(k\frac{\partial T}{\partial z}\right) + \mathcal{L} \tag{5}$$

in which $c$ is the specific heat capacity of the firn, $G$ the diffusive ground heat flux, $k$ the thermal conductivity of the firn and $\mathcal{L}$ a heat source representing the release of latent heat upon the refreezing of liquid water in the firn or the subsurface absorption of solar radiation. Subsurface penetration of short-wave radiation is neglected in the current model version. This is deemed a reasonable approximation for fine-grained, polar snow (Van Dalum et al. (2020)); Brandt and Warren (1993) showed that most of the absorption of light occurs in the IR and in the first few centimetres of the snow. The firn temperature profile
is initialized using a spin-up period, see Sect. 2.1.5. Before the spin-up, the firn column is initialised at a constant temperature equal to the annual mean surface temperature during the spin-up period. The lower boundary condition assumes a constant heat flux across the lowest model grid cell, i.e. the deep temperature is allowed to change along with long-term changes in surface temperature or internal heat release. The upper boundary condition for the temperature calculation is provided by the surface ('skin') temperature in RACMO2, obtained by iteratively solving the surface energy balance (Van Den Broeke et al. (2008)).
Due to the Lagrangian character of the model, vertical heat advection is implicitly considered (Helsen et al. (2008)). Any heat generated by firn horizontal/vertical deformation is neglected.

The heat equation is solved numerically using the so called "splitting method". In the first half of a time step we solve for water transport using the bucket-scheme (described in more detail in the Sect. 2.1.4). Temperature changes caused by the

refreezing of meltwater are added as a source term. Then, in the second half of the time step, no water flux is allowed, making every layer a closed system. We then allow heat conduction to take place by solving Eq. 5 implicitly using the Crank-Nicolson scheme.

The thermal conductivity is assumed to depend on firn density and temperature, and in previous versions of IMAU-FDM followed the expression for seasonal snow due to Anderson (1976), which only depends on density. In the updated model, the parameterization for thermal conductivity as a function of firn density of Calonne et al. (2019) is used. The new expression more accurately models the dynamics of the thermal conductivity by incorporating both a density and temperature dependency. The new expression was obtained from 3D images of firn micro-structures at different temperatures, and is valid for the wide range of density and temperature values typically encountered in ice sheet firn layers, making it suitable for simulations of the GrIS. It takes on the following form:

$$k(\rho, T) = (1 - \theta) \frac{k_i(T) k_a(T)}{k_i(-3\circ\mathrm{C}) k_a(-3\circ\mathrm{C})} k_{snow}(\rho) + \theta \frac{k_i(T)}{k_i(-3\circ\mathrm{C})} k_{firn}(\rho) \tag{6}$$

The equation consists of two parts: one for snow and low-density firn and one for ice and high-density firn. The transition between the two regimes remains smooth through the weight factor $\theta(\rho)$. The definition of $\theta$ and the thermal conductivities that are used in Eq. 6 are:

$$\theta = 1/(1 + \exp(-0.04(\rho - 450)))$$

$$k_i(T) = 9.828 \exp(-0.0057T)$$

$$k_a(T) = (2.334 \cdot 10^{-3} T^{3/2})/(164.54 + T)$$

$$k_{snow}(\rho) = 0.024 - 1.23 \cdot 10^{-4} \rho + 2.5 \cdot 10^{-6} \rho^2$$

$$k_{firn}(\rho) = 2.107 + 0.003618(\rho - \rho_i)$$

Here $k_a$ represents the thermal conductivity of air, taken from Reid et al. (1959). Figure 3 compares the old and new expressions for various temperatures. As can be seen in Fig. 3, the new expression takes on a slightly lower value than Anderson (1976) at densities below $\sim 475 - 565 \ \mathrm{kg \ m^{-3}}$, depending on the temperature, but a higher value at densities above that. This difference becomes larger at lower temperatures.

### 2.1.4 Meltwater percolation, retention and refreezing

IMAU-FDM employs a tipping bucket method to treat the percolation, irreducible (capillary) retention and (re)freezing of water, by filling up subsequent deeper layers to maximum capacity in a single model time step (i.e. quasi-instantaneous). Magnusson et al. (2015) show that, in spite of its simplicity and shortcomings, the tipping bucket method is a robust and useful method to deal with liquid water transport in the snowpack when compared to more sophisticated methods, especially when capturing general firn properties at the larger, multi-kilometre horizontal scale for which IMAU-FDM is designed. In IMAU-

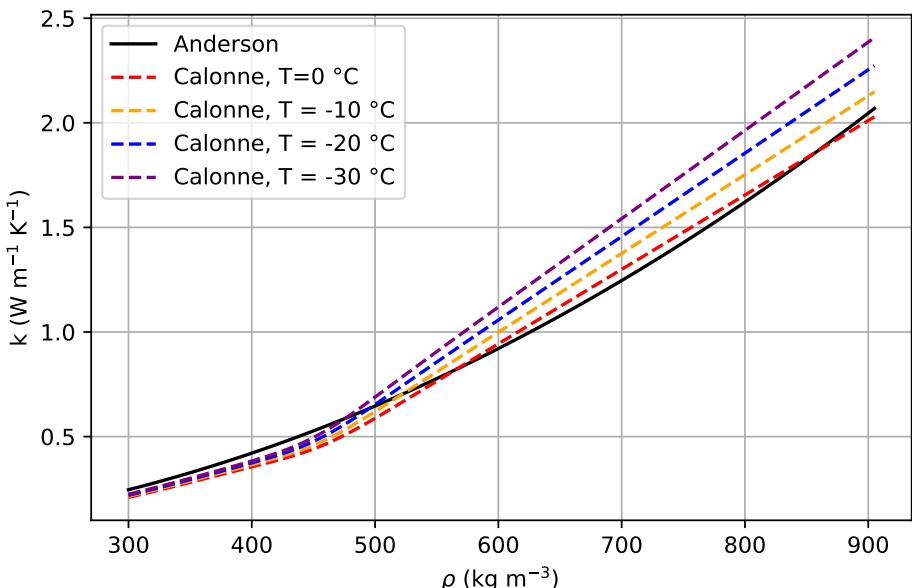

**Figure 3.** Comparison of the thermal conductivity parameterisation by Anderson (1976) and Calonne et al. (2019) with density at different temperatures.

FDM, the fraction that is retained in a model layer by capillary forces ('irreducible water content') depends on the available pore space according to the expression by Coléou and Lesaffre (1998):

$$W_c = 1.7 + 5.7 \frac{P}{1 - P} \tag{7}$$

where $W_c$ is the irreducable water content in percents of volume and $P$ is the porosity of the firn layer, defined as $P = 1 - \rho/\rho_i$. The maximum amount of water that is stored thus decreases with increasing density of the firn layer. Standing water and as a consequence lateral runoff over ice-layers are currently ignored. This is considered a fair assumption, because on the spatial scales at which the model is employed (i.e. the RACMO2 grid of $5.5$ by $5.5$ km) it is assumed that within a model grid cell the meltwater can usually find a way to flow around a layer of ice. While ponding of meltwater plays a role in the formation of ice layers, IMAU-FDM is still able to simulate ice layers. However, the ice layers usually consist of multiple thin layers of high density ice, often interlaid with a thin layers with slightly lower density. An example of a location with ice slabs is KAN-U. A figure showing the old and new modelled density profile of KAN-U can be found in the supplementary material.

### 2.1.5 Model initialisation

The latest IMAU-FDM model runs span the period 10 November 1957 - 31 December 2020. The initial model density, temperature and liquid water content in the firn column are obtained by repeatedly applying the spin-up period 1960 - 1979

during which the forcing (i.e. surface accumulation, liquid water flux and temperature) is assumed to have remained reasonably constant (i.e. no significant long-term trends, Van Den Broeke et al. (2009)). Observations and model studies support the assumption that the Greenland climate and SMB started to change significantly in the 1990s (Enderlin et al. (2014); McMillan

et al. (2016)), confirming that the period 1960 - 1979 can be selected for initialization purposes. Initialization is considered complete when the entire firn layer (up to the pore close-off density of $830 \text{ kg m}^{-3}$) has been refreshed. The required number of iterations depends on accumulation rate, and is typically $25$ to $45$ for the relatively dry grid points in the northeastern GrIS and typically $10$ to $20$ for the relatively wet southeastern GrIS. After the spin-up is finished, the actual model run starts by applying the 1957-2020 forcing from RACMO2.3p2 once.

## 2.2  RACMO2.3p2 forcing

At the upper boundary of IMAU-FDM, mass accumulation (solid precipitation minus sublimation minus drifting snow erosion), liquid water fluxes (melt plus rainfall minus evaporation) and surface temperature are prescribed from the regional atmospheric climate model RACMO2.3p2, which has been used to simulate the climate and surface mass balance of the GrIS and its immediate surroundings for the period 1958-2020 at a horizontal resolution of $5.5 \text{ km}$. Like previous versions, this version of

RACMO2 has been extensively evaluated over the GrIS (Noël et al. (2018b)). At the lateral boundaries, using a relaxation zone of 24 gridpoints, RACMO2 is forced by European Centre for Medium-Range Weather Forecasts (ECMWF) re-analysis data, i.e. ERA-40 between November 1957 and 1978, ERA-Interim between 1979 and 1990 and ERA-5 between 1991 and 2020. For the forcing of IMAU-FDM the full spatial resolution of $5.5 \text{ km}$ is used and a temporal resolution of 3 hours was selected, as an acceptable trade-off between robustly resolving the daily cycle and keeping manageable file sizes. IMAU-FDM typically

uses a timestep of $15 \text{ min}$, for which we linearly interpolate the forcing between the RACMO2 forcing time steps. Both v1.1G and v1.2G have been forced by the version of RACMO at the same resolution. In this way, all the differences between the two versions that are presented in this paper are due to the difference in the model physics.

### 2.3  Firn thickness and elevation change

IMAU-FDM tracks the total firn thickness, and changes in it. The resulting vertical velocity of the ice-sheet surface due to

changes in the firn layer ($\frac{\mathrm{d}h}{\mathrm{d}t}$) is given by:

$$\frac{\mathrm{d}h}{\mathrm{d}t} = v_{snow} + v_{snd} + v_{er} + v_{melt} + v_{ice} + v_{fc} \tag{8}$$

The total vertical surface velocity $\frac{\mathrm{d}h}{\mathrm{d}t}$ can thus be decomposed into separate contributions from accumulation ($v_{snow}$), surface sublimation (included in $v_{snow}$), sublimation by snowdrift ($v_{snd}$), erosion or deposition by snowdrift ($v_{er}$), snowmelt ($v_{melt}$), and firn compaction ($v_{fc}$). The term $v_{ice}$ is defined as the mean surface mass balance (SMB, $v_{ice} = v_{snow} + v_{snd} + v_{er} + v_{melt}$)

with an opposite sign. It represents the long-term average vertical mass flux through the lower boundary of the firn column, which equals the mass flux through the upper boundary in a steady-state firn layer. In Sect. 4 we show surface elevation change and the individual components for three case study locations on the GrIS.

## 2.4 Observations

IMAU-FDM output is evaluated using previously available and newly obtained profiles of firn density, temperature and liquid water content from the GrIS accumulation zone. In total there are 124 observations, which cover a wide area to ensure that the various ice facies and climate zones of the GrIS are well represented (Fig. 4). These observation consist of 92 firn cores, 31 observations of the temperature at 10 metre depth and one GPR observation of meltwater intrusion. Vertical profiles of observed firn density from ice cores vary in depth from $9.6$ to $150.8$ m and have been drilled between 1952 and 2018. The cores come from various sources, such as the Program for Arctic Regional Climate Assessment (PARCA; McConnell et al. (2000); Mosley-Thompson et al. (2001); Hanna et al. (2006); Banta and McConnell (2007)), the Arctic Circle Traverses (ACT, Box et al. (2013)) and the EGIG line (Harper et al. (2012)), Das 1 and Das 2 (e.g. from Hanna et al. (2006)) and several other cores were retrieved from the SUMup database (SUrface Mass balance and snow depth on sea ice working grouP), (Koenig et al. (2013); Montgomery et al. (2018)). Of these cores, 29 are used for calibration (see Sect. 2.1.2). All the cores are used for evaluation in Sect. 3.1. Table S1 in the supplementary material lists all cores that have been used, their coordinates, the year in which it has been drilled, depth and corresponding citation.

Temperature observations include profiles ranging in depth between $4$ and $14$ m obtained by Harper et al. (2012) along a transect in the western GrIS and at the NEEM deep ice core drilling site (Orsi et al. (2017)). Additional firn temperature observations are from Summit, Dye-2 (Steffen et al. (1996) as processed by Vandecrux et al. (2020a), KAN-U (Charalampidis et al. (2015)) and FA-13 (Koenig et al. (2014)). An additional 14 observations of $10$ m firn temperatures are from Polashenski et al. (2014). More data concerning these observations, such as their coordinates and year of retrieval, is listed in Table S2 of the supplementary material.

For observations of liquid water in firn, we use observations from Dye-2 (Heilig et al. (2018)), obtained using an upward-looking ground-penetrating radar (upGPR), which was installed and operated in the summer of 2016. The upGPR was buried $\sim 4.5$ m under the snow, and was capable of measuring the liquid water percolation depth, content as well as the changing distance between the instrument and the snow surface. Although the data do not cover a wide spatial (single location) or temporal range (1 May-16 October 2016), they are unique and moreover have high temporal and vertical resolution, making them very valuable for firn model evaluation (Vandecrux et al. (2020b)), but also to evaluate melt intensity and timing in the forcing time series.

# 3 Model performance

## 3.1 Firn density

The vertical density profiles of 92 GrIS firn cores are used to assess the performance of the updated model. For each available firn core, IMAU-FDM has been run at the grid point closest to that location. The evaluation is not completely independent of the calibration, as the 29 cores used for fitting the MO-values are also included. As an integrated measure of porosity, we compare modelled and observed vertically integrated firn air content (FAC), i.e. the vertical distance over which the firn layer

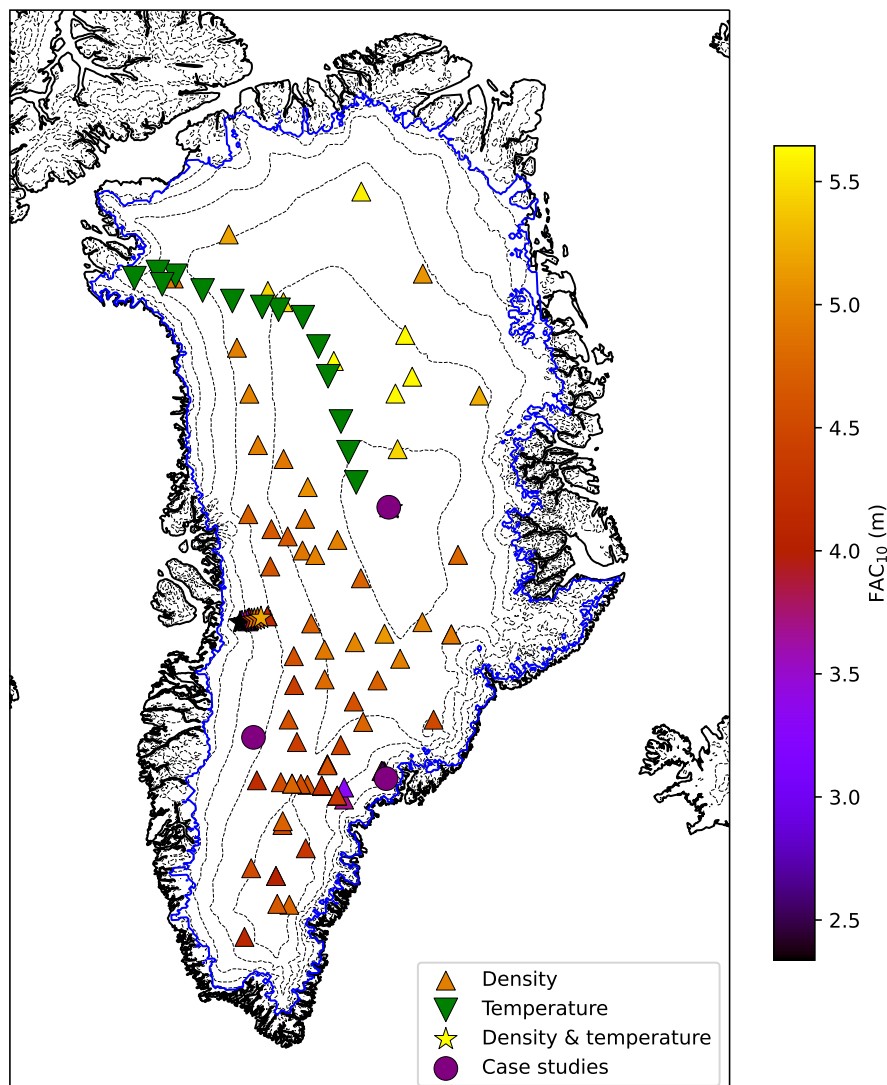

**Figure 4.** Locations of observed density (upward triangle), 10 m temperature (downward triangle), both (stars). The colour of the upward triangles and stars indicate the measured firn air content for the first 10 m of snow at that location (FAC$_{10}$). The three purple circles indicate the locations of the case studies discussed in Sect. 4. Dashed lines represent 500 m elevation contours, the blue solid line the contiguous ice sheet margin. The name of each location is listed in the supplementary material.

can be compressed until reaching the density of glacier ice across the entire firn column. FAC is an indicator of the meltwater retention capacity of the firn layer and therewith an important parameter to simulate correctly.

$$\text{FAC} = \sum_{j}^{n_z} \frac{(\rho_i - \rho_j)}{\rho_i} \Delta z_j \tag{9}$$

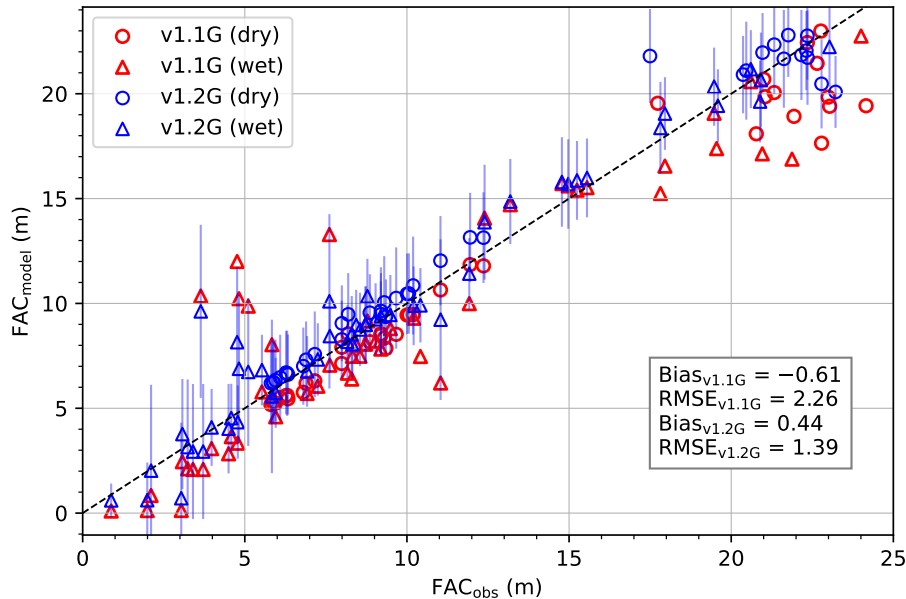

**Figure 5.** Modelled vs observed firn air content in metres. Dry locations are indicated with circles whereas wet locations are indicated with triangles. A location is labelled as dry if it experiences 5% less melt than accumulation during the spin-up period. The blue lines indicate the uncertainty in the v1.2G results.

Here, $n_z$ is the number of layers in that firn profile, $\Delta z_j$ is the thickness of layer $j$ and $\rho_j$ is the density of that layer. The FAC is calculated over the depth range at which observations are available. This means that if, for example, a core goes to a depth of $20$ m then the FAC is calculated over the top $20$ m, but if a different core is $40$ m deep then the FAC for that data point FAC is calculated over the top $40$ m of firn. Note that this is different from $FAC_{10}$ shown in Fig. 4, which was calculated over the top $10$ m. This ensures that we compare the FAC with observations over the largest depth range possible. In general, the more melt a location on the GrIS experiences the lower its FAC, and shorter firn cores also lead to a lower FAC.

With the newly adopted parametrizations, the simulation of FAC in dry locations has significantly improved (Fig. 5). The mean difference between the modelled and observed FAC (bias) has decreased from $0.61$ to $0.41$ m. Simultaneously, the root of the mean square of the difference between the modelled and observed FAC (RMSE) of all cores has decreased from $2.16$ to $1.39$ m. There are 39 locations with FAC $> 15$ m, which are all relatively dry. For these locations, the root mean squared error (RMSE) decreased from $1.68$ to $1.11$ m (-34%). The improvement is more modest for low FAC locations, where the previous underestimation has now become a small overestimation. This can be attributed to the new fresh snow density parameterization, which results in lower densities especially close to the surface. For these 53 locations, the RMSE decreased from $2.60$ to $1.57$ m (-40%). Eq. (2) leads to a lower surface density, which in turn leads to a higher FAC. Simultaneously, a higher densification rate leads to a lower FAC, which suggests that the improvement in the modelled FAC stems mostly from the new fresh snow density parametrization, whereas the new densification rate ensures that the firn profile is modelled correctly at greater depths.

As mentioned in Sect. 2.1.1, Fausto et al. (2018) suggested using a constant density of $315 \text{ kg m}^{-3}$ instead of using a temperature dependent formulation. Here, we analyse the model's performance when using a constant fresh snow density of $315 \text{ kg m}^{-3}$ and compare that to the model's performance when using Eq. 2. We do this by comparing the RMSE in the FAC. Moreover, we define a core-specific RMSE in firn density, $\Phi$, as an additional metric to quantify the error in the modelled vertical density profile:

$$\Phi = \sqrt{\frac{1}{L_z} \sum_{i}^{n_z} \left(\rho_{model,i} - \rho_{obs,i}\right)^2} \tag{10}$$

With $L_z$ the length of the core. This error quantifies the RMSE in the profile when comparing the modelled profile to the observations. The mean $\Phi$ then quantifies the mean performance of the model, and gives a slightly different result than the RMSE in the FAC. The RMSE in FAC decreased from 2.26 m in v1.1G, to 1.39 m in v1.2G, where the surface snow density depends on annual mean temperatures. When employing a constant density of of $315 \text{ kg m}^{-3}$ the RMSE is slightly higher: 1.44 m. Similarly, the mean $\Phi$ of all density profiles decreased from $2.1 \cdot 10^3 \text{ kg m}^{-3}$ to $2.0 \cdot 10^3 \text{ kg m}^{-3}$ when using surface snow density that depends on annual mean temperatures and $1.9 \cdot 10^3 \text{ kg m}^{-3}$ when using a constant density respectively. These results show that including temperature as a predictor for the surface density does improve model performance, but only for FAC, not for $\Phi$. Thus, we opt for the temperature dependent formulation, since this also allows capturing the effect of long-term temperature trends on the surface density.

Fig. 6 shows observed and modelled density profiles at Das 2 and FA-13, sites with large and small FAC respectively. Das 2 is a dry location, with very little melt ($4 \text{ mm w.e.yr}^{-1}$) and changes to its profile whereas FA-13 experiences strong melt ($406 \text{ mm w.e.yr}^{-1}$). The aquifer site was selected because its facies represent a distinct climatological zone on the GrIS, with a combination of high melt and high accumulation, which we expect will results in distinct firn characteristics. Standing water is not allowed in IMAU-FDM v1.2G, while this is known to occur at firn aquifer sites (Koenig et al. (2014); Miège et al. (2016); Montgomery et al. (2017); Miller et al. (2020)), so that modelled liquid water content remains a qualitative rather than quantitative estimate. In spite of this, it has previously been shown that the model accurately reproduces the spatial distribution of aquifers (Forster et al. (2014)), from which we conclude that first order processes that determine temperature and pore space (FAC) are sufficiently represented. At both sites, the changes made in v1.2G improved the modelled density-depth profile, with a more realistic shape and reduced variability. It increases the pore space and brings simulated FAC in better agreement with the observed density profile. For both locations, the modelled FAC has improved: the bias at Das 2 has decreased from $+1.3$ m to $-1.0$ m and at FA-13 the bias has decreased from $-2.2$ to $+0.3$ m. Simultaneously, the $\Phi$ at both locations has remained nearly identical: an increase of 2.2 and 0.1 % at Das 2 and FA-13 respectively. One of the reasons for the increased performance is the change to a surface density parametrization that is based on annual mean temperature values instead of the temperature at the time of the snow event. This leads to greatly reduced "peaks" in the density profile, which is more in line with observations.

For FA-13, the lower surface density also matches the upper 25 m of the density profile better. Despite the improvement, the densification rate in the upper region is still too high. This may be attributed to the lack of a description of microstructural

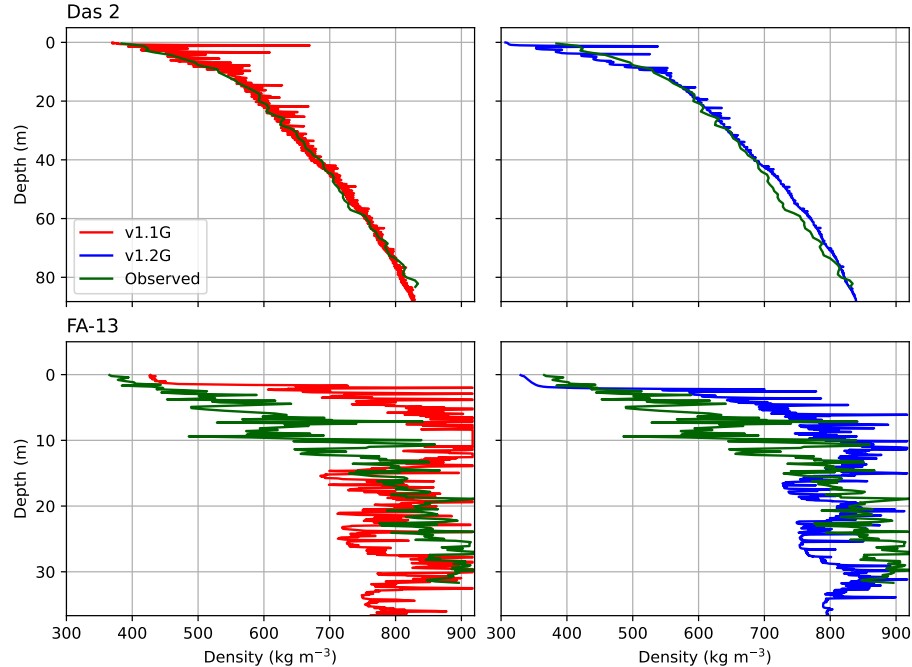

**Figure 6.** Density profiles for v1.2G (left) and v1.1G (right) model settings at Das 2 (top) and FA-13 (bottom).

properties on the firn. In the presence of liquid water the rate at which snow grains grow is increased. Firn with larger grains lead to a lower densification rate. This feedback is currently not present in the model. The presence of liquid water may also reduce the densification rate in a different way: it reduces the effective stress felt by the firn layer, which is the driving force for densification. This process is often observed in soils, where it is called consolidation: initially water takes up a change in stress before the soil matrix. To our knowledge, however, the influence of the pore water pressure on the effective stress has not been investigated in the context of firn densification.

### 3.2 Firn temperature

Modelled and measured $10\,\mathrm{m}$ firn temperatures at 31 locations are compared in Fig. 7. We make the distinction between relatively cold locations ($T_{10} < -20\,°\mathrm{C}$) and warm locations ($T_{10} > -20\,°\mathrm{C}$). Model version 1.2G performs better than v1.1G, especially for the warmer locations with significant melt, which are mostly locations from Harper et al. (2012) in West Greenland. The error at these warmer locations has been significantly reduced: for locations with $T_{10} > -20\,°\mathrm{C}$, the mean RMSE decreased from 4.7 to 3.1 °C, respectively (-43%). A better representation of the density at those locations allows for more realistic refreezing and the associated enhanced latent heat release, increasing the temperature in melt-prone locations. Additionally, the lower conductivity due to the lower density leads to a less efficient cooling of the deeper snow during winter. In spite of the clear improvement, a cold-bias remains in IMAU-FDM for some of these locations. On the other hand, for the

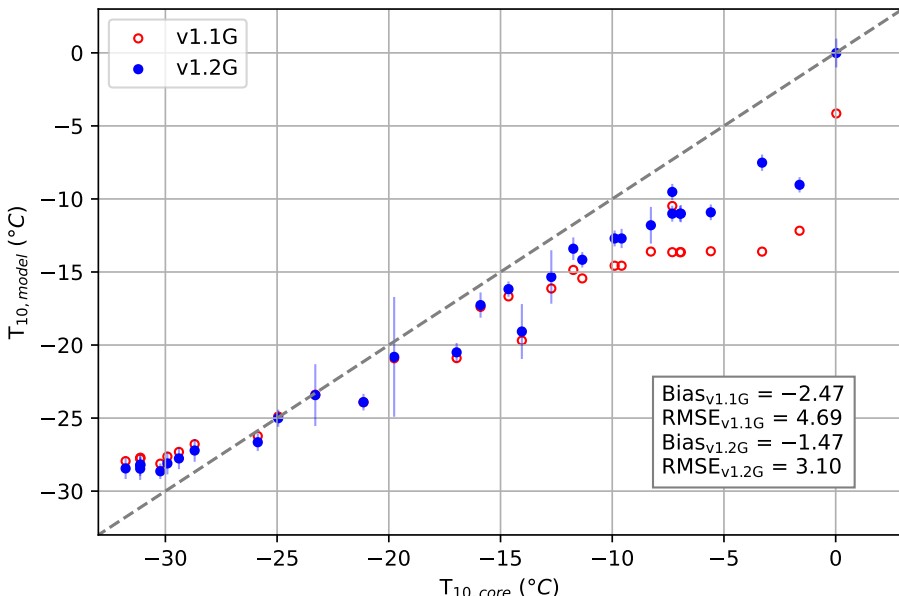

**Figure 7.** Modelled vs. observed temperature at 10 m depth (in °C) for 31 locations on the GrIS.

cold, low-melt locations ($T_{10} < -20\,°C$), a persistent warm bias in IMAU-FDM remains. It appears that the bias in $T_{10}$ varies spatially. At the relatively dry and cold locations, located mostly in the interior and North of the GrIS, the remaining temperature bias can come from uncertainty in the RACMO forcing (surface temperature, liquid water input, snow accumulation). This is opposed to the low-lying wet areas, where the uncertainties in the firn model (heat conduction, meltwater percolation, pore space availability, depth of refreezing). The model also does not include firn ventilation, which can warm or cool the firn depending on the season (Albert and Shultz (2002); Steger et al. (2017b)). Further research is needed to clarify the exact reasons for these remaining biases.

Fig. 8 compares the observed temperature profiles of Summit and Dye-2 in winter and summer with the results from IMAU-FDM v1.1G and v1.2G. Similarly to what was found in Fig. 7, Summit, which is a dry and cold location has a warm bias whereas Dye-2, which is relatively mild and wet, has a cold bias. We deem the differences with observations of 1 to 2 °C at these locations acceptable in light of the potential uncertainties in both forcing and firn processes, as described above. At both locations, we see that the temperature gradients have gotten larger near the surface. This indicates that heat diffuses slower in the upper layers. For both locations the new surface density parameterization has decreased the density in the upper layers. This in turn leads to a lower thermal conductivity since the thermal conductivity increases monotonically with density (Fig. 3). Furthermore, Eq. 6 leads to lower values for the thermal conductivity than the previously used parametrization by Anderson (1976) at densities $\sim < 500\,\mathrm{kg\,m^{-3}}$.

The summer profile of Dye-2 clearly shows a temperature maximum in v1.2G. Such a maximum was not present in v1.1G and is also not present at Summit. It is found that refreezing occurs at a greater depth than before, see Sect. 3.3, which

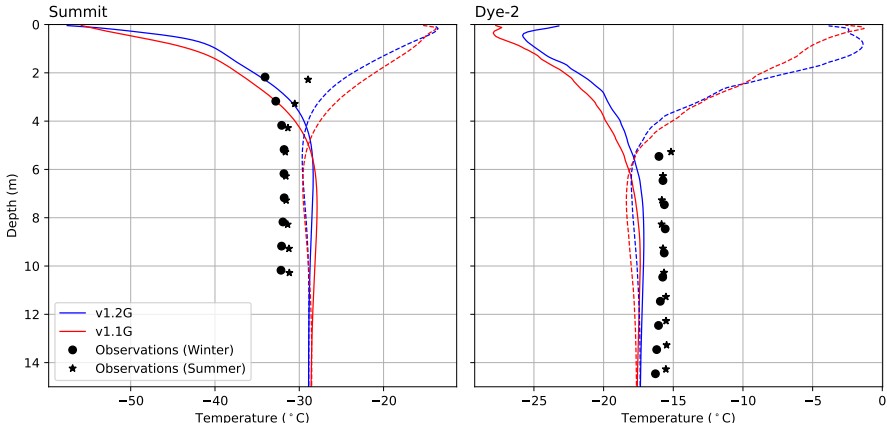

**Figure 8.** Comparison between observed temperature profiles vs. modelled profiles by v1.1G and v1.2G in summer (dashed lines) and winter (solid lines) at Summit in winter (9 March 2002) and summer (6 August 2002) and Dye-2 in the summer (10 August 2007) and winter (13 March 2007). The blue lines indicate the uncertainty in the v1.2G results.

corresponds to the depth at which we see the temperature maximum. It is possible that refreezing of meltwater heated up the firn at Dye-2 earlier in Summer, which would explain why we see a temperature maximum at Dye-2 but not at Summit. Diffusion will transport the heat released by refreezing towards greater depths and the surface and eventually the local peak in temperature caused by the refreezing will disappear. However, in v1.2G the rate of diffusion has decreased. This slower rate of diffusion, combined with refreezing occurring a greater depth than before, causes the elevated temperatures to persist for a
longer period of time, which may explain why it is visible in v1.2G but not in v1.1G.

### 3.3 Liquid water content

The liquid water percolation and retention schemes have not been updated, but the changes made to the parameterizations that impact density and temperature do influence water percolation, and therewith liquid water content (LWC), and these changes are discussed here. Very few in-situ, vertically resolved observations of LWC are available. Here we used data from a recent
study that used upward looking ground penetrating radar (upGPR) at Dye-2 in the higher percolation zone of the southwestern GrIS ($2120 \, \mathrm{ma.s.l.}$, see Fig. 4, Heilig et al. (2020)). The observations have an hourly temporal resolution.

Fig. 9 compares the results of v1.1G and v1.2G against the observed evolution of the maximum penetration depth and LWC in the firn. The measurements reveal that the melt in 2016 at Dye-2 mostly occurred in four periods between July and October, the timings of which are well captured in the RACMO2.3p2 forcing. Comparing v1.1G and v1.2G, the water penetration depth
and LWC have both increased. This mainly reflects the decreased density in the upper layers at Dye-2 (see the density profile shown in the supplementary material). As discussed in the previous section, this leads to an increase in the temperature in the upper firn layer and stronger temperature gradients at Dye-2. The increase in temperature means that the water needs to percolate deeper into the firn pack before it can refreeze, which is reflected in the increased penetration depth. Simultaneously,

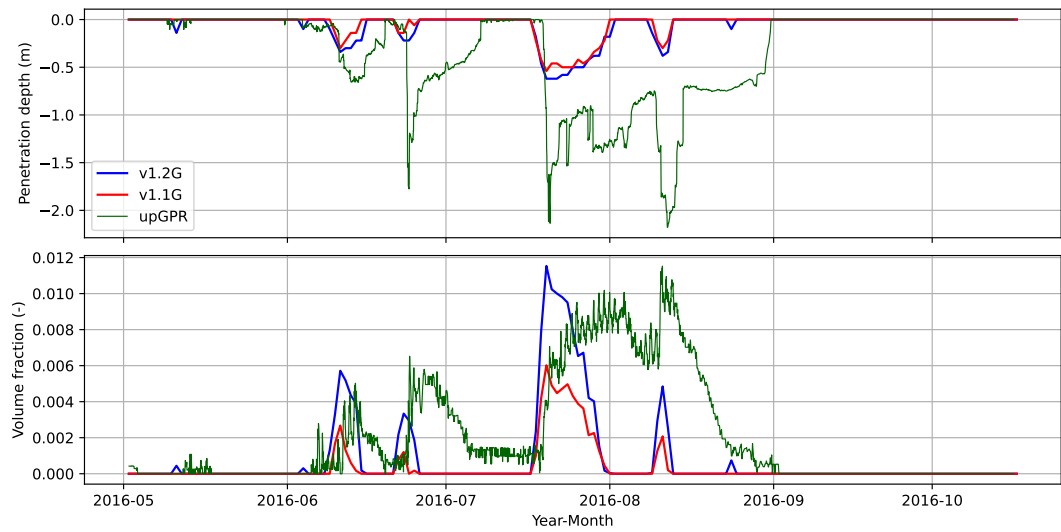

**Figure 9.** Comparison between the observed penetration depth (top) and volume fraction (bottom) of liquid water at Dye-2 with the results from v1.1G and v1.2G.

the decrease of the surface density means that there is more pore space near the surface that can retain irreducable water, explaining the increase in volume fraction. Overall, the penetration depth now agrees better with the observations, although in IMAU-FDM, the meltwater still refreezes too quickly in more shallow layers than observed. The RMSE of the penetration depth has decreased from $0.54$ to $0.52\,\mathrm{m}$. The RMSE is calculated over the whole period over which data is available, including periods during which no liquid water was detected. Similarly, the RMSE in the volume fraction has decreased from $3.41 \cdot 10^{-3}$ to $3.33 \cdot 10^{-3}$. Varying the irreducible water content by, e.g., multiplying Eq. 7 with a constant factor or using a constant volume or mass fraction, did not improve the result, and it was decided to leave the liquid water scheme unchanged. The remaining deficiencies between the modelled and observed l.w.c. are in part due to still too low temperatures at Dye-2 compared to the measurements despite the improvements. This means that the liquid water will reach firn freezing point at a shallower depth. As mentioned in the previous section, remaining errors in the temperature profile come from a combination of uncertainties in the forcing (surface temperature, liquid water input, snow accumulation, surface density) and uncertainties in the firn model (heat conduction, meltwater percolation, pore space availability, depth of refreezing).

## 4 Case studies

In this section we compare time series (1958-2020) of firn-induced surface elevation (i.e. firn depth) changes at three key locations: Summit in the cold and dry ice sheet interior, KAN-U in the relatively warm and dry southwestern percolation zone and FA-13 in the wet and relatively mild southeastern firn aquifer region (Koenig et al. (2014); Forster et al. (2014), as indicated

**Table 2.** Location and climate climate of the three case study sites. The annual mean accumulation are calculated over the whole simulation period (1957-2020).

| | Lon. (°W) | Lat. (°N) | Elevation (m a.s.l.) | $T_{2m}$ (°C) | Acc. (mm w.e.) | Melt (mm w.e.) |
|---|---|---|---|---|---|---|
| **Summit** | 38.32 | 72.55 | 3281 | -26.0 | 206 | 0 |
| **KAN-U** | 47.02 | 67.00 | 1840 | -12.4 | 480 | 271 |
| **FA-13** | 39.04 | 66.18 | 1563 | -7.0 | 986 | 496 |

by the purple circles in Fig. 4). Table 2 provides geographical and climatological information of these locations. The density profiles of Summit and KAN-U are given in the supplementary material.

## 4.1 Summit

Summit is located at the centre of the GrIS at a high elevation and therefore it experiences a low amount of snowfall and a negligible amount of rain and melt. The evolution of its elevation is therefore closely linked to changes in the temperature

(higher temperatures lead to a higher compaction rate) and accumulation (higher accumulation leads to a thicker firn layer). Panels a and c in Fig. 10 show how annual accumulation and mean annual skin temperature change over the course of the simulation period, as well as changes to the surface elevation and its velocity components, both for v1.1G and v1.2G.

At Summit, an 0.8 m elevation change between 1970 and 2000 is modelled (which equals $\sim 2.5$ cm yr$^{-1}$ when divided by the number of years), with a slightly decreasing firn depth in the periods before and after. This can be explained by lower

accumulation before about 1970 and after 2000, along with slightly increased temperatures since 2000. Differences in simulated surface elevation between v1.1G and v1.2G are small, in spite of the individual velocity components being different. As shown in panel g of Fig. 10, the interannual variability in firn depth is dominated by snowfall ($v_{snow}$), which is compensated mainly by steady firn compaction ($v_{fc}$). From this it follows that the slightly higher accumulation and lower temperature between $\sim$1970 and 2000 caused the net upward surface velocity. Overall, the net vertical velocity of the surface is very similar between v1.1G

and v1.2G, because both $v_{snow}$ and $v_{fc}$ have increased in magnitude almost equally. The new surface density parameterization (Eq. 1) leads to a lower surface density ($\sim 60$ kg m$^{-3}$ lower at Summit), which in turn is compensated for by a higher densification rate in order to match the observed set of $z_{550}$ and $z_{830}$ values. Thus, with a lower surface density, the vertical upward velocity of the surface is higher, and the compaction velocity $v_{fc}$ has also increase by an almost equal amount ($\sim 0.18$ m yr$^{-1}$). This explains why overall the total vertical velocity and the resulting surface elevation change does not differ much

between v1.1G and v1.2G.

However, v1.2G does show larger seasonal and interannual oscillations in the firn depth. This is because $v_{snow}$ and $v_{fc}$ act on different timescales. $v_{fc}$ is fairly constant in time and changes in tandem with the seasonal changes in temperature. $v_{snow}$ on the other hand is much more variable, as snowfall is very episodic and highly variable at multiple time scales. This also implies that the firn model is now more sensitive to changes in the forcing, reacting more strongly to changes in accumulation

and skin temperature.

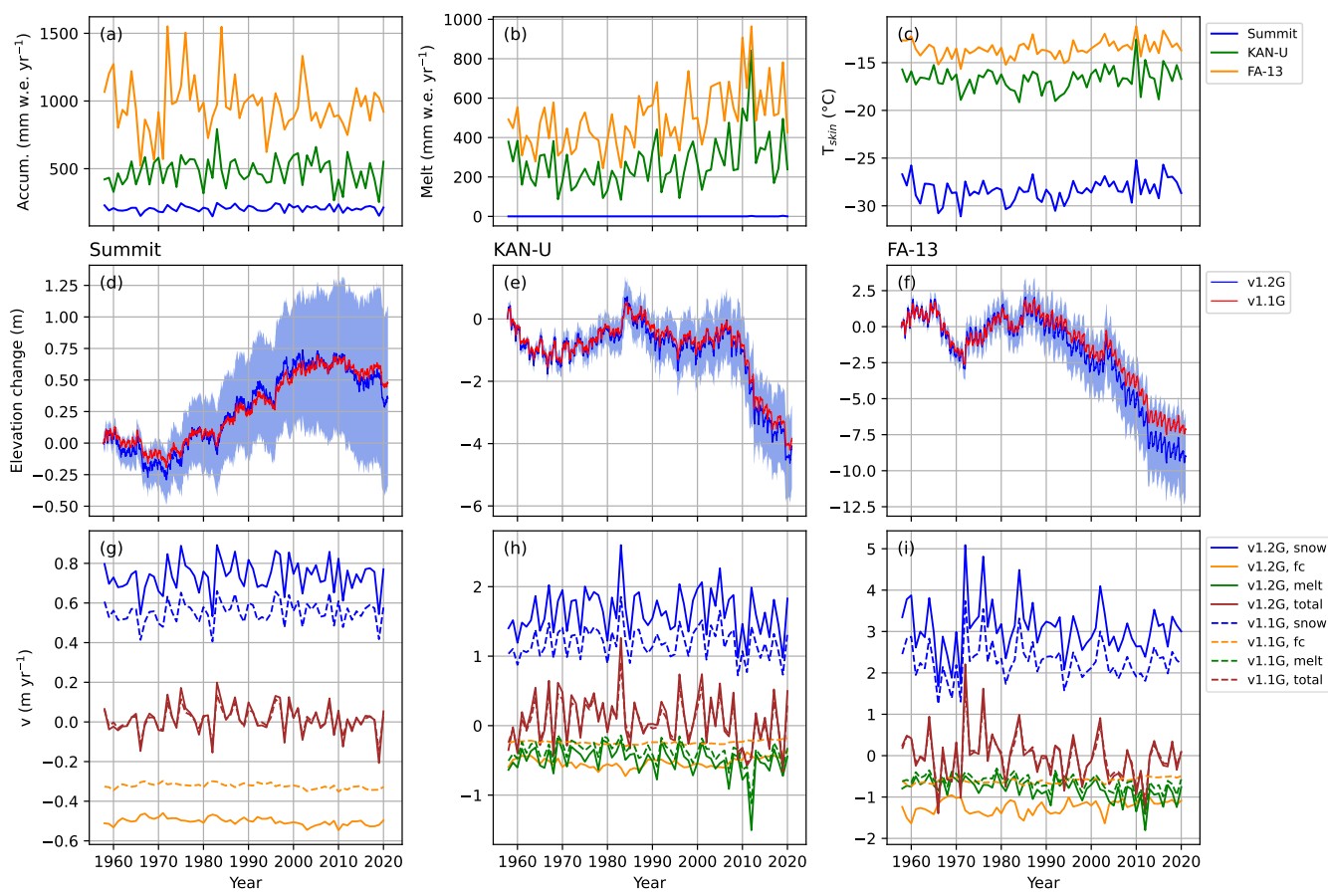

**Figure 10.** Time series of the total annual accumulation (a), total annual melt (b) and the annual mean skin temperature (c) at Summit, KAN-U and FA-13. Panels (d)-(f) show the net cumulative surface elevation change since the start of the simulation for these three locations. The blue area indicates the uncertainty in the v1.2G results. Finally, panels (g)-(j) show the mean annual velocity components that cause the elevation change.

## 4.2 KAN-U

Situated in the southwest and at a lower elevation, KAN-U is warmer than Summit and melting occurs every year during the summer, which greatly affects the firn properties at its location (Fig. 10b). The average influence of surface melt on firn depth changes ($v_{melt}$, Fig. 10h) is similar in magnitude and sign to the contribution from compaction ($v_{fc}$): it decreases the depth of the firn column and decreases its air content. At KAN-U, a 2.5 m thickening is modelled between 1970 and 1985. If we look at the associated climate forcing, this can be explained by a relatively low amount of melt and a low temperature during this period. Between 2005 and 2020, a rapid decline ($-20$ cm yr$^{-1}$) is observed, which is associated with increased surface melt since 2005, as well as a slight increase in temperature and a reduction in accumulation. The most striking changes in elevation at KAN-U occurred in exceptional years; for example, 1983 was a very wet year, 2010 was a warm year, and 2012 was a high-melt year.

When comparing the vertical surface velocities between v1.1G and v1.2G, we again see an increased accumulation velocity $v_{snow}$, due to a lower fresh snow density, compensated for by a more negative compaction velocity resulting in a similar net velocity ($v_{tot}$). As accumulation decreased after 2005, the net effect is a slight surface lowering. Following significant warming and increased melt at this site (Fig. 10b, c), the contribution of $v_{melt}$ to firn depth changes increases and that of $v_{fc}$ decreases, making the former the dominant process leading to surface lowering at KAN-U. $v_{melt}$ is also larger in magnitude in v1.2G because the melted snow at the surface is of a lower density, even though the amount of mass that melts is unchanged.

## 4.3 FA-13

FA-13, a site with a firn aquifer, experiences a warmer and wetter climate than KAN-U which leads to a rapid densification in the upper part of the firn column (Fig. 6). Here, the signal is dominated by large seasonal oscillations in firn depth of up to $\sim 1$ m yr$^{-1}$ between 1960 and 1985. From 1985 onwards, the firn depth decreases until 2012 at a higher rate in the updated than in the previous model ($\sim 0.35$ vs. 0.25) m yr$^{-1}$.

For the vertical velocity components, a similar picture emerges at FA-13 as at KAN-U: $v_{melt}$ becomes the dominant source of elevation lowering since 2005. The contribution of melt to surface elevation variations also becomes more important, because the melt itself increases and becomes more variable while the variability in accumulation decreases over time. At FA-13, the compaction is stronger and shows more interannual variability, in line with the larger interannual variability of the annual accumulation.

Both the variability and the magnitude of the melt is stronger in v1.2G. In the period 1990-2020, 8.5 m of thinning occurred in v1.2G, compared to 6 m in v1.1G. Since the uppermost layers of snow are structurally less dense in v1.2G, enhanced surface melt implies a stronger lowering of the surface, especially in strong melt years (10i).

## 5 Uncertainty analysis

In order to quantify the model uncertainty, we performed sensitivity tests in which the model settings, spin-up settings and the RACMO forcing were varied. The parametrizations for the snow density (Eq. 2) and the thermal conductivity (Eq. 6) were varied by one standard error: $44 \mathrm{\ kg\ m^{-3}}$ (Fausto et al. (2018)) and $0.05 \mathrm{\ W\ K^{-1}\ m^{-1}}$ (Calonne et al. (2019)). Following a similar procedure as Kuipers Munneke et al. (2015b), accumulation and melt has been increased or decreased by 10%. Finally, the temperature was varied by $0.5\ ^{\circ}\mathrm{C}$. Only one variable was changed at a time. An overview of the conducted tests is given in Table 3. The spread in the results is indicative of the sensitivity and accuracy of the model results. The variance and covariance of each test were combined quadratically to obtain a value for the total uncertainty. The resulting uncertainties are shown in Fig. 5 and 7 as error bars and in Fig. 10 as a shaded band around the elevation change signal.

These error margins do not include the uncertainty caused by missing physical processes in the model. For example, the lack of deep water percolation may cause additional errors at wet locations. From these tests, it turns out that the uncertainty in the modelled FAC (Fig. 5), is mainly caused by uncertainties in the surface snow density, skin temperature and accumulation. However, the relative importance varies per location: wetter and warmer locations are more sensitive to the forcing.

The sensitivity of the modelled $T_{10}$ (Fig. 7), scales almost one-to-one with the uncertainty imposed on the temperature during the spin-up period and during the simulation period. However, for some locations such as KAN-U (located around $-20$ $^{\circ}\mathrm{C}$), the uncertainty is much larger, because melt at these locations turns out to be especially susceptible to a slightly higher skin temperature. Simultaneously, $T_{10}$ at these locations is very susceptible to a small increase in the amount of meltwater. This combination makes these error bars larger. Remaining biases in the results are most likely due to missing physics in the handling of liquid water percolation.

The blue shaded area in Fig. 10 shows the uncertainty in the firn depth at Summit, KAN-U and FA-13. This uncertainty grows almost linearly over time. At Summit, the cumulative uncertainty at the end of 2020 is in the same order of magnitude as the signal itself. These uncertainties are dominated by uncertainties during the spin-up period. We found that the elevation change is sensitive to the initialised firn profile and spinning up the model with a slightly altered climate will result in a drift in the elevation change. We used the period 1960-1980 as a reference climate during the spin-up. However, this climate is not exactly representative of the firn's history and this thus introduces an error. Hawley et al. (2020) found that the mean surface velocity at Summit equals $1.9 \mathrm{\ cm\ yr^{-1}}$ during the period 2008-2018. IMAU-FDM v1.2G reports a small decrease of $0.2 \mathrm{\ cm\ yr^{-1}}$, with an uncertainty of $3.2 \mathrm{\ cm\ yr^{-1}}$, indicating no significant trend. For the wetter and warmer locations KAN-U and FA-13, the errors are larger in the absolute sense, but relatively smaller.

## 6 Summary and outlook

Temporal and spatial variability in firn layer thickness is highly relevant for studying the mass balance of the Greenland ice sheet (GrIS), because it directly impacts its refreezing efficiency. Moreover, firn thickness change is an important component of surface elevation change, and improved knowledge is required to accurately convert remotely sensed GrIS volume to mass changes. In this paper, we presented improvements in the offline version of the IMAU firn densification model (IMAU-FDM

**Table 3.** List of sensitivity tests conducted for determining the model uncertainty.

| Test # | Variable | Variation | When |
|---|---|---|---|
| 1 | $T_{skin}$ | $\pm0.5\,^\circ\mathrm{C}$ | Spin-up |
| 2 | $\dot{b}$ | $\pm10\%$ | Spin-up |
| 3 | Melt | $\pm10\%$ | Spin-up |
| 4 | $T_{skin}$ | $\pm0.5\,^\circ\mathrm{C}$ | Whole run |
| 5 | $\dot{b}$ | $\pm10\%$ | Whole run |
| 6 | Melt | $\pm10\%$ | Whole run |
| 7 | $\rho_0$ | $\pm44\,\mathrm{kg\,m^{-3}}$ | Whole run |
| 8 | $k$ | $\pm0.05\,\mathrm{W\,K^{-1}\,m^{-1}}$ | Whole run |

v1.2G), forced by three-hourly output of the regional climate model RACMO2.3p2. Taking advantage of improved climate forcing and newly available observations of surface and subsurface firn density and temperature, the improvements are systematically implemented in the parametrizations of surface density, dry snow densification and thermal conductivity. The treatment of liquid water is not changed, owing to a lack of sufficient observations to justify changes in the current configuration.

The updated model predicts higher firn air content (FAC), which at three selected sites in the interior GrIS and in the southwestern and southeastern percolation zone results in a larger sensitivity of firn thickness to intra- and interannual variations in snowfall, melt and temperature. As an important consequence of a change in fresh snow density parameterization, the inter- and intra-annual variations in elevation have increased, owing to an increased sensitivity to changes in its forcing. In a warmer climate, firn thinning owing to increased surface melt becomes increasingly important at the marginal sites, both in the mean and as a component of interannual variability. Future applications of the improved model include a full GrIS assessment of contemporary and future firn mass and thickness changes, as well as explaining areas where firn aquifers and ice slabs currently occur, and their future changes.

*Code availability.* The code of IMAU-FDM v1.2G used in this project is available on GitHub at https://github.com/brils001/IMAU-FDM and at Zenodo (https://zenodo.org/record/5172513, Brils et al. (2021)).

*Data availability.* J. E. Box, E. Mosley-Thompson, J. R. McConnell, K. Steffen, J. T. Harper and S. B. Das provided us with some of the firn core data that has been used to calibrate and validate IMAU-FDM. The rest of the firn cores were obtained from the SUMup dataset (Montgomery et al. (2018)). A list of all firn cores used and corresponding references can be found in the Supplementary Material. Firn temperatures are obtained from Vandecrux (2020), Polashenski et al. (2014) and Harper et al. (2012). The upGPR data of the liquid water measurements at Dye-2 are available from Heilig et al. (2018).

*Author contributions.* MB, PKM, WJvdB and MRvdB started this project, decided on its scope, which parts of the model required further development and interpreted the results. MB performed the model simulations, implemented the changes to the model, comparisons and led the writing of the manuscript. All authors contributed to discussions on the manuscript.

*Competing interests.* The authors declare that they have no conflict of interest.

*Acknowledgements.* This work was carried out under the program of the Netherlands Earth System Science Centre (NESSC), financially supported by the Ministry of Education, Culture and Science (OCW grant no. 024.002.001). We acknowledge ECMWF for computational time on their supercomputers.

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
