# Peer review of "Improved representation of the contemporary Greenland ice sheet firn layer by IMAU-FDM v1.2G"

_Geoscientific Model Development, 2021_

## Referee Comment (RC3)

Review of
*Improved representation of the contemporary Greenland ice sheet firn layer by IMAU-FDM v1.2G*
by Brils et al.

Reviewer: C. Max Stevens

This paper details updates to the commonly used IMAU firn densification model. It demonstrates the new model's improvements by comparing model outputs from the new and old IMAU models using several metrics. The paper is clearly written and well organized. The science and ideas presented are well founded. The paper will be a good contribution to *Geophysical Model Developments*, and I am happy to recommend its publication after the authors address several questions I have.
* * *
**General Comments:**

1. My impression after reading the paper was that the authors focus almost entirely on the improvements made in the new model and do not discuss uncertainties and limitations of the model. Given that firn models are often cited as the (or one of the) largest contributors of uncertainty to altimetry-derived mass balance estimates and the fact that numerous research groups use IMAU-FDM, I think it would be useful to include a section discussing the uncertainties in the model's outputs. Additionally, beyond just providing uncertainty bounds (e.g. +/-X%) I think it would be useful in that section to include discussion of why these uncertainties persist – is it due to missing representations of physical processes in the model, the way the model is calibrated, a propagation of uncertainties in the forcing data, or something else?

2. I was surprised to see the firn aquifer site chosen as a test site because the authors state explicitly (line 209) that their model does not simulate standing water on ice layers, which is what is occurring in the firn aquifer zone. Why should one expect the model to perform well at a site where it is not configured to simulate the observed conditions?

3. I am curious about why the authors only use cores from dry sites (line 153) to derive the densification model – this seems fine, but then they use the equation derived for dry sites to simulate firn compaction in the wet-firn zone. (Or, line 153 states, "MO corrects for the dry compaction rate" – is there an additional factor added to correct for wet compaction rate?) Why is using a dry-firn equation for wet firn a valid thing to do? Is there good reason to believe that an equation developed for dry-firn compaction should simulate wet-firn compaction equally well? Why not use cores from wet-firn sites to make a more general MO?

4. I would like more detail about the latent heat source term $L$ in equation 5. How is that implemented in practice? What is the numerical scheme you are using? How do you determine what $L$ is? E.g., are you using an enthalpy solving method, solving the heat equation for temperature and then making a correction for layers where there is liquid water with temperature below freezing?

5. I would like more discussion of why you chose surface density to be a function of the previous year's temperature. Wouldn't you expect surface density to change on short time scales based on local conditions (e.g. warm snow event, cold snow event, strong wind saltation)? You are running your model at very high temporal resolution, which to me means that you think your equations adequately describe physical processes well enough to predict how the density evolves at sub-daily timescales. But, it seems that you are saying that you can only predict your boundary condition at annual timescales – why should I believe that the 3-hour resolution density profiles (or elevation change predictions) are correct when the surface boundary is not calculated with that resolution? Please provide some discussion on this apparent paradox. Is the reality that our knowledge of what determines surface density is at present deficient, and so the best course of action is to not introduce additional model uncertainty by using an equation that does not work well?

6. "old" vs. "new" settings: several times in reading the paper I got confused reading about old vs new. Your new model is v1.2G – what was the old one called – v1.1G? I suggest referring to them throughout the paper by version number rather than "old" and "new", and reference your equations when needed. An example of where confusion arises: on line 122, are you saying that "old" and "new" only refer to which density parameterization you are using, or does it refer to the updated densification equation also?

7. Equation 3: Does $\dot{b}$ evolve through time based on changing accumulation rates through time, or is it constant for a given site based on the spin up climate? If the latter, why? Imagine a site that has warmed since 1980; why should the densification rate of snow that fell in e.g. 2010 be dependent on the accumulation rate from decades earlier when the climate was different? Arthern et al. (2010) formulated their model equation using a steady-accumulation assumption (see their Appendix B); is it appropriate to use your densification equation with variable accumulation, either climatic variability or changing climate? Why not just formulate your model in terms of the stress?
* * *
**Line by line comments:**

Introduction – your introduction is very much about melt, but the rest of your paper is more about the firn densification process and your new equations to simulate that (and your meltwater scheme is unchanged from previous versions). Additionally, your pilot application (section 4) is about altimetry and not melt processes. I suggest editing the introduction to talk a bit more broadly about firn and specifically include a bit more information about altimetry. As it is currently, I read the first paragraph and thought the paper was going to be about the firn's decreased ability to retain meltwater and threshold behavior associated with that; it isn't until the 4th paragraph that you get to topics that are more specifically addressed in your paper.

Line 15: firn doesn't "represent" the transitional stage, it **is** the transitional stage

L22: 'collapses' – choose a different word to differentiate between collapse meaning is damaged – perhaps something like "until at some point the pore space is insufficient to accommodate melt and the system is fundamentally changed"

L24-25: much more extensive than what?

L29: do you mean reduces *by* a factor of 1-4 (also, why the large range on that factor?)

L30: just say "at least 44%"; "no less than" can confuse the reader

L56: describes → describe

L58: resulted → result

L69: I believe that SUMup specifies that each core used should be cited with its original publication to give full credit to those to worked to generate those data, rather than broadly just citing the SUMup papers. I have seen this done by including a supplement listing the cores used and the appropriate citation for each.

L90: remove sentence here about subsurface radiation – you say that again later, which is a more appropriate place for that information

Figure 1 – caption says purple circles, but they are green. Text later in paper (L325) says green. I suggest switching to purple to differentiate between sites and temperature measurements.

L108/Eq. 1: Fausto et al. (2018) concluded that using a value of 315 kg m$^{-3}$ is better than using a parameterization equation. You do mention this later in your paper, but I think it would be better to move some of that text to this point in the paper and reference the results/sensitivity tests in Section 3.1.

Table 2/Figure 3/Line 155 – You state the R$^2$ value is 3e-3 – perhaps I am misinterpreting what you are doing, but that number to me indicates a very poor fit. Perhaps expand in your text what you are regressing? You should also specify in Table 2 that $\sigma$ is the variable you are using to notate standard error. And, what is the standard error you are calculating – does this imply you have some normal distribution you are looking at? I am not sure what that statistic is telling me or how you are calculating it.

L159: use caution with word "below" – do you mean deeper than that density horizon in the firn, or densities less than 550? (likewise "above").

L168: I am confused – you say first on 168 that $L$ includes subsurface absorption of radiation, but then on the next line you say you ignore it.

L202 – citation formatting error

L209: You state that the latter assumption is thought to be valid; what about the first (standing water)?

219: missing a *v* for snowdrift – should be $v_{snd}$ to be consistent with others

L245: Can you explain the 3-minute time stepping in more detail? I am not sure what you mean here. Do you run the model at 3-minute resolution and then just save the results every 3 hours?

256: "here the FAC is calculated" Do you mean that Equation 9 calculates the FAC over the entirety of the firn column?

258: FAC<15 melt – is this an observation from you model results? Say so.

L261-265: I am confused – the numbers you write in this paragraph are different than those shown in Figure 5.

Figure 5 and L264: define what 'Bias' means specifically, i.e. how are you calculating it, and what is it actually a measure of?

L261-265: You should be more consistent delineating 'FAC in dry' vs 'low FAC' either use wet/dry or high/low, and state your thresholds

263: why did it switch from underestimation to overestimation?

L267: here is another example of confusion with old/new – in this case you are just referring to the old/new densification equations?

Eq. 10: Isn't this the RMSE? Why are you referring to it as cost function and calling it Phi, and then pivoting in line 271 and calling it RMSE (and then again using Phi at line 273)?

L271-276: Is the improvement of v1.2G over the "old" model, as demonstrated in Figures 5 and 6, due to the new densification equation, or due to the improvement of the surface density parameterization?

L293: I would expect that a cold bias would be more due to incorrect handling of meltwater (e.g. not enough refreezing, so not warm enough) rather than due to a cold bias in the RACMO forcing, especially given that cold sites have a warm bias. Can you comment on this? Regarding the warm bias, does this mean RACMO is biased warm in the colder areas, despite you saying that RACMO is biased cold in the previous sentence?

L295/Figure 8: Indeed, the new model does slightly better than the old, but it appears to be an incremental improvement, and the bigger issue is that there is a still a substantial misfit compared to the observations (more than 5 degrees at 2 m), especially at Summit in the summer. Please provide discussion on this misfit, and the implications it has on your other results (i.e., a temperature difference of 5 degrees will substantially alter the densification rate predicted by your densification equations).

301: Why do you not include shallower observations of temperature, which are available for DYE-2? Wouldn't this provide an additional metric to test how well the model is capturing the latent heat release due to refreezing?

301: How much of the difference in modeled temperature is due to the (a) new conductivity parameterization, (b) the new surface density, and (c) the new densification equation?

319: Figure 9 shows that your model is predicting penetration depth that is too shallow by a factor of 3 – but your text focuses on the improvement of the new model over the old. Please add text describing (or hypothesizing) the remaining deficiencies in the model that cause it to fail to predict the penetration depth accurately.

335: Did you fit trendline to the 1970 to 2000 time series to find the m/yr increase, or are you just differencing the 2000 and 1970 values, and then dividing by the number of years?

338: "Nevertheless, the individual velocity components being very different" – sentence structure issue – "are very different"?

Figure 10: Your model predicts that the elevation has lowered in the last decade, but surface elevation measurements from Summit (Hawley et al., 2020; sorry to self-reference but it is the dataset I am most familiar with) show that the elevation at Summit increased at 0.019 m/year from 2008 to 2018. Can you explain your model's inconsistency with the observations?

Figure 10: I am curious about the sudden decrease in elevation in 2019-2020, which appears to be related to the low accumulation in 2019. Is that decrease consistent with altimetry data?

340: Given that $v_{snow}$ and $v_{fc}$ offset each other, is your new model effectively the same from an altimetry standpoint as the old model? By this I mean: is the new formulation a better mathematical representation of physical processes occurring in nature? Or, does 'new' differ from 'old' just in that you added several cores to the calibration and changed the surface density, so the calibration coefficients are different? Does this mean that in using your densification equation, the surface density must be prescribed as you do with Equation 2? If so, can your equation be used to model firn in a location where Equation 2 is not valid?

References:
Hawley, R. L., Neumann, T. A., Stevens, C. M., Brunt, K. M., & Sutterly, T. C. (2020). Greenland Ice Sheet elevation change: Direct observation of process and attribution at summit. *Geophysical Research Letters*, *47*, e2020GL088864. https:// doi.org/10.1029/2020GL088864

---

## Author Comment (AC1)

◊ Baptiste Vandercrux

o Xavier Fettweis

• Max Stevens

We would like to thank the editor and reviewers for their detailed and constructive comments. Please find below our actions and responses, grouped by comment type and section. The reviewer's name corresponds to the symbol used, see above.

**General comments:**

◊ A **better acknowledgment of previous work** is required throughout the manuscript. Lastly, the lack of attention to the citations and to the reference list made the reading irritating on occasions.

We agree that more attention to the references is needed. By including the suggested references (see detailed comments below) as well as some other citations within the text we hope to have remedied this.

• My impression after reading the paper was that the authors focus almost entirely on the improvements made in the new model and do not discuss uncertainties and limitations of the model. Given that firn models are often cited as the (or one of the) largest contributors of uncertainty to altimetry-derived mass balance estimates and the fact that numerous research groups use IMAU-FDM, I think it would be useful to **include a section discussing the uncertainties in the model's outputs**. Additionally, beyond just providing uncertainty bounds (e.g. +/-X%) I think it would be useful in that section to include **discussion of why these uncertainties persist** – is it due to missing representations of physical processes in the model, the way the model is calibrated, a propagation of uncertainties in the forcing data, or something else?

This is a good point. In the revised manuscript, we will include in the discussion section a brief discourse on the remaining uncertainties, specifically addressing the role of model uncertainties versus spin-up and forcing uncertainties. This will be done along the lines of Kuipers Munneke et al., 2015.

• "old" vs. "new" settings: several times in reading the paper I got confused reading about old vs new. Your new model is v1.2G – what was the old one called – v1.1G? I suggest referring to them throughout the paper by version number rather than "old" and "new", and reference your equations when needed. An example of where confusion arises: on line 122, are you saying that "old" and "new" only refer to which density parameterization you are using, or does it refer to the updated densification equation also?

We agree. To avoid confusion, the manuscript has been changed to always refer to the old and new version of the model as v1.1G and v1.2G respectively, and to reserve the words "new" and "old" to refer only to a specific change in a model parametrization, such as for surface density.

**Line by line comments**

**1: Introduction**

- Introduction – your introduction is very much about melt, but the rest of your paper is more about the firn densification process and your new equations to simulate that (and your meltwater scheme is unchanged from previous versions). Additionally, your pilot application (section 4) is about altimetry and not melt processes. **I suggest editing the introduction to talk a bit more broadly about firn and specifically include a bit more information about altimetry.** As it is currently, I read the first paragraph and thought the paper was going to be about the firn's decreased ability to retain meltwater and threshold behavior associated with that; it isn't until the 4th paragraph that you get to topics that are more specifically addressed in your paper.

An important application of firn models is a realistic simulation of meltwater retention properties. Another important application is converting altimetry measurements to a mass balance estimate. As the reviewer rightfully notes the latter has not received enough attention in our introduction, which focuses mostly on the first application. To remedy this, we have added a paragraph dedicated to altimetry between the first and second paragraph. The other paragraphs have been reworded slightly to accommodate for this change.

- ◊ I also realize that the introduction misses some background about the use of firn models in altimetry to fully motivate this firn height analysis. Reference to previous work on firn thickness should be acknowledged, e.g.:

  Sørensen, L. S., Simonsen, S. B., Nielsen, K., Lucas-Picher, P., Spada, G., Adalgeirsdottir, G., Forsberg, R. and Hvidberg, C. S.: Mass balance of the Greenland ice sheet (2003-2008) from ICESat data -The impact of interpolation, sampling and firn density, Cryosphere, 5(1), 173–186, doi:10.5194/ tc-5-173-2011, 2011.

  Zwally, H. Jay, and Li Jun. "Seasonal and interannual variations of firn densification and ice-sheet surface elevation at the Greenland summit." Journal of Glaciology 48, no. 161 (2002): 199-207. Li, J. and Zwally, H.J., 2011. Modeling of firn compaction for estimating ice-sheet mass change from observed ice-sheet elevation change. Annals of Glaciology, 52(59), pp.1-7.

  Hawley, R. L., Neumann, T. A., Stevens, C. M., Brunt, K. M., & Sutterley, T. C. (2020). Greenland Ice Sheet Elevation Change: Direct Observation of Process and Attribution at Summit. Geophysical Research Letters, 47(22), e2020GL088864.

Thank you. These citations have been incorporated into the newly added paragraph, among other citations.

- Line 15: firn doesn't "represent" the transitional stage, it is the transitional stage

This sentence has been reworded and the word "represent" has been removed.

◊ l.20: Please acknowledge other work about on meltwater refreezing in the Canadian Arctic and in Greenlandic peripheral glaciers. e.g.

Gascon, G., Sharp, M., Burgess, D., Bezeau, P., & Bush, A. B. (2013). Changes in accumulation-area firn stratigraphy and meltwater flow during a period of climate warming: Devon Ice Cap, Nunavut, Canada. Journal of Geophysical Research: Earth Surface, 118(4), 2380-2391.

Bezeau, P., Sharp, M., Burgess, D., & Gascon, G. (2013). Firn profile changes in response to extreme 21st-century melting at Devon Ice Cap, Nunavut, Canada. Journal of Glaciology, 59(217), 981-991. Ashmore, D. W., Mair, D. W., & Burgess, D. O. (2020).

Meltwater percolation, impermeable layer formation and runoff buffering on Devon Ice Cap, Canada. Journal of Glaciology, 66(255), 61-73 .

Thank you. These citations have been added to the text.

• L22: 'collapses' – choose a different word to differentiate between collapse meaning is damaged - perhaps something like "until at some point the pore space is insufficient to accommodate melt and the system is fundamentally changed"

This sentence has been reworded to better explain that the pore space is being filled up and not being mechanically damaged.

◊ l.24: Please acknowledge other work about on meltwater refreezing in Greenland, e.g.:

Pfeffer, W. Tad, Mark F. Meier, and Tissa H. Illangasekare. "Retention of Greenland runoff by refreezing: implications for projected future sea level change." Journal of Geophysical Research: Oceans 96.C12 (1991): 22117-22124.

Braithwaite, R. J., Pfeffer, W. T., Blatter, H., & Humphrey, N. F. (1992). Meltwater refreezing in the accumulation area of the Greenland ice sheet: P  kitsoq, summer 1991. Rapport Grlands Geologiske Underselse, 155, 13-17.

Thank you, we have now added these citations to the introduction.

• L24-25: much more extensive than what?

This referred to smaller peripheral ice caps and glaciers in Greenland. In order to avoid confusion, the adjectives "much more" have been removed.

• L29: do you mean reduces by a factor of 1-4 (also, why the large range on that factor?)

This number comes from the study by Harper at al. (2012), who estimated a lower and an upper bound to the total amount of meltwater that the pore space in the percolation zone can take up. In the study, they measured the available pore space along a transect in southwest Greenland and extrapolated these results to arrive at an estimate for the whole Greenland ice sheet. The uncertainty stems from fitting a power law through their observations. The sentence in the manuscript has now been changed to reflect this, i.e.: "is reduced by a factor ~1-4".

- L30: just say "at least 44%"; "no less than" can confuse the reader

This has been changed according to the reviewer's suggestion.

◊ l.39: "Zwally…" These two first references are for Antarctica. Please use work done in Greenland.

This is a good point. We will add citations of work done in Greenland.

- L56: describes → describe

This has been changed.

- L58: resulted → result

This has been changed.

◊ l.59: Please add a quick sentence for Section 3

Section 2.2 was incorrectly referenced instead of Section 3. This has been changed and the sentence has been reworded slightly for clarity.

**2.1: Observations**

◊ l.63: Here you describe how the model output is evaluated before you describe the model and the output. I recommend changing the structure to describe the model first (which should be at the center of the GMD article). Please make the distinction between the observations that are used to improve the model and the ones that are used for evaluation.

We agree with the reviewer that it is preferable to first discuss the model before the observations. Therefore, we have moved this part of the text to the end of Section 2. To make clearer which cores are used, where they were drilled and where they are used in the paper, we have created a table that sums up this information as a supplement. In this table we labelled observations that have been used to derive the new MO-fit.

◊ l.70: From the SUMup ReadMe: "When using this dataset please cite both the individual researchers who provided the data as listed in the Citation column as well as the SUMup dataset."

Missing citations are now included in the table mentioned above.

◊ l.73: please cite "Steffen et al. 1996 as processed by Vandecrux et al. 2019"

Steffen, C., Box, J., and Abdalati, W.: Greenland Climate Network: GC-Net., CRREL Special Report on Glaciers, Ice Sheets and Volcanoes, trib. to M. Meier, 96, 98–103, 1996.

The reference has been changed accordingly.

- L69: I believe that SUMup specifies that each core used should be cited with its original publication to give full credit to those to worked to generate those data, rather than broadly just citing the SUMup papers. I have seen this done by including a supplement listing the cores used and the appropriate citation for each.

As suggested by the reviewer, we have created a supplement that lists each core, the correct citation and in which figures of the paper this core is used.

**2.2: IMAU-FDM**

- L90: remove sentence here about subsurface radiation – you say that again later, which is a more appropriate place for that information

As suggested, this sentence has been removed.

- Figure 1 – caption says purple circles, but they are green. Text later in paper (L325) says green. I suggest switching to purple to differentiate between sites and temperature measurements.

The figure has been updated such that the circles are purple. L325 has been changed accordingly.

**2.2.1: Fresh snow density**

- General comment: I would like more discussion of **why you chose surface density to be a function of the previous year's temperature**. Wouldn't you expect surface density to change on short time scales based on local conditions (e.g. warm snow event, cold snow event, strong wind saltation)? You are running your model at very high temporal resolution, which to me means that you think your equations adequately describe physical processes well enough to predict how the density evolves at sub-daily timescales. But, it seems that you are saying that you can only predict your boundary condition at annual timescales – **why should I believe that the 3-hour resolution density profiles (or elevation change predictions) are correct when the surface boundary is not calculated with that resolution?** Please provide some discussion on this apparent paradox. Is the reality that our knowledge of what

determines surface density is at present deficient, and so the best course of action is to not introduce additional model uncertainty by using an equation that does not work well?

Thank you for this observation. While it is true that the actual density of fresh snow varies on much shorter time scales than our parameterization, we have opted here for a parameterization that depends on annual mean surface temperatures. There are two reasons for this approach. Firstly, the parameterization is derived by fitting the measured snow densities to mean **annual** temperatures, not the temperature at the time of the accumulation event. Thus, the equation itself links snow density to annual temperatures, not instantaneous temperatures. Therefore, using the instantaneous temperatures would introduce an additional uncertainty. Secondly, in deriving their parameterization, Fausto et al. (2018) used the density of the upper 10 cm of snow. Especially in low accumulation locations, this means that the measured layer of firn contains snow from multiple accumulation events. Moreover, it may also have compacted in the time between the accumulation event and the observation. Therefore, we believe that the typical temperature to which this 10 cm of snow is exposed to can more reasonably be approximated with annual temperatures than with instantaneous ones. The text of the manuscript has been updated to reflect our argument.

- L108/Eq. 1: Fausto et al. (2018) concluded that using a value of 315 kg m-3 is better than using a parameterization equation. You do mention this later in your paper, but I think it would be better to move some of that text to this point in the paper and reference the results/sensitivity tests in Section 3.1.

We think it is best to leave the discussion of the model performance to Section 3. Therefore, we decided to not move text of the model's performance. Instead, we included a new paragraph here that foreshadows these results since we do think it is appropriate to already mention Fausto's conclusion in this part of the text.

**2.2.2: Dry snow densification rate**

- Equation 3: **Does $\dot{b}$ evolve through time based on changing accumulation rates through time, or is it constant for a given site based on the spin up climate?** If the latter, why? Imagine a site that has warmed since 1980; why should the densification rate of snow that fell in e.g. 2010 be dependent on the accumulation rate from decades earlier when the climate was different? Arthern et al. (2010) formulated their model equation using a steady-accumulation assumption (see their Appendix B); is it appropriate to use your densification equation with variable accumulation, either climatic variability or changing climate? **Why not just formulate your model in terms of the stress?**

Here, we take b as a constant. It is equal to the mean annual accumulation during the spin-up period (1960-1980), as this period is the climate in which the firn layer is assumed to be in equilibrium. The actual stress $\sigma(t)$ that a firn layer of age $t_{age}$ "feels" (in the absence of melt) is given by:

$$\sigma(t_{age}) = g \int_0^{t_{age}} \dot{b}(t') \, dt'$$

Taking $\dot{b}(t)$ as a constant is an approximation that simplifies the densification rate, but the uncertainty this introduces is minor. This can be demonstrated by considering a hypothetical layer of firn that is deposited right after the spin-up ends at Summit. Now we can compare the stress of on this layer if we use a constant $\dot{b}$ or calculate the stress using the equation above. The evolution of the stress is shown in the figure below:

[Figure]

The error grows over time. At the end of 2020 the error is equal to 3.2%. For other locations we obtain similar results (e.g. for Dye-2 the error is 1.9%). This error is small enough to justify this approach for the timescale considered. However, we acknowledge that for larger time scales, such as for future runs, this approach may no longer be valid. Therefore, we plan to make appropriate adjustments to the model before doing such runs. We have added a short paragraph to the revised manuscript to summarize our findings presented above.

- General comment: I am curious about why the authors only use cores from dry sites (line 153) to derive the densification model – this seems fine, but then they use the equation derived for dry sites to simulate firn compaction in the wet-firn zone. (Or, line 153 states, "MO corrects for the dry compaction rate" – **is there an additional factor added to correct for wet compaction rate?)**

Eq. 3 describes the firn densification resulting from overburden pressure. This equation is used for both wet and dry locations, and so it is assumed that the densification rate (due to overburden pressure) of dry firn is equal to that of wet firn. We acknowledge that the presence of liquid water in between grains may impact the evolution of their size and shape. This in turn may also impact the compaction rate of the firn. To our knowledge, most firn models that account for a different densification rate of wetted firn are based of Vionnet et al., 2012. They introduce this dependency through an empirical correction factor for the firn viscosity. This correction factor is derived from experiments that have not been published (see Brun et al., 1992). Due to a lack of physical understanding and a lack of available measurements we decided not to introduce an extra dependence of the compaction rate on the liquid water content to reduce the model's complexity and to prevent overfitting. We have added a short paragraph to the revised manuscript to summarize these considerations.

- General comment: **Why is using a dry-firn equation for wet firn a valid thing to do?** Is there good reason to believe that an equation developed for dry-firn compaction should simulate wet-firn compaction equally well? Why not use cores from wet-firn sites to make a more general MO?

Our formulation using MO is a semi-empirical correction to the compaction rate (Eq. 3). MO is shorthand for Modelled over Observed. This is the ratio between the modelled and observed depth at which the density profile reaches 550 kg/m3 in the case of $MO_{550}$ or 830 kg/m3 in the case of $MO_{830}$. This ratio is found to correlate with accumulation which is the reason why we opt for this approach (see also Ligtenberg et al., 2011). We purposefully do not incorporate wet-firn sites into our derivation of the fit. This is because at wet locations, the density at a given depth is not only impacted through compaction but also through the vertical transport of liquid water inside of the firn after a melt event. This drastically alters the depth at which the firn layer reaches 550 or 830 kg/m3. Incorporating these ratios of depths into the derivation of the MO fit would incorrectly assign this change to a different compaction rate whereas it is due to refreezing. To make this clearer, we have added a short paragraph to the revised manuscript to summarize these considerations.

◊ Table 1: old values for alpha and beta do not match with the values given line 143-144. Do they come from different studies?

These values indeed come from a different study, namely Ligtenberg et al. (2011). Since the values from Kuipers et al. (2015) are already mentioned in Table 1 we have removed these from the text to avoid confusion and repetition.

- Table 2/Figure 3/Line 155 – You state the R2 value is 3e-3 – perhaps I am misinterpreting what you are doing, but that number to me indicates a very poor fit. Perhaps expand in your text what you are regressing? You should also specify in Table 2 that $\sigma$ is the variable you are using to notate standard error. And, what is the standard error you are calculating – does this imply you have some normal distribution you are looking at? I am not sure what that statistic is telling me or how you are calculating it.

Ligtenberg et al. (2011) found that the ratio of the depth at which the model reaches 550 kg/m3 and the depth at which the observations reach 550 kg/m3 (MO) are correlated to the mean annual accumulation at that site. This correlation is described well by Eq. 4. The same goes for the depth at which the firn density reaches 830 kg/m3. Here we rederive this fit using more observations and a different model and model forcing. Now, we find that correlation between MO and accumulation has almost disappeared for 550 kg/m3. For 830 kg/m3, the two are still strongly correlated, as indicated by the R2 of both regressions. $\sigma$ denotes the standard error of the estimators ($\alpha$ and $\beta$). We calculate this standard error to quantify the uncertainty of the estimators; we can be 95% certain that the value of the estimator lies within one standard error of its value. The standard error is calculated by assuming that the errors in the regression are normally distributed (the normality assumption). It can then be calculated via:

$$se = \sqrt{\frac{\frac{1}{n-2}\sum_{i=1}^{n} e_i^2}{\sum_{i=1}^{n}(x_i - \bar{x})^2}}$$

Where n is the degrees of freedom, $e_i = y_i - \bar{y}$ are the residuals (the difference between the samples and the sample mean), and x is the predictor. We have added a sentence to the revised text to explain how we arrived at the uncertainty.

- L159: use caution with word "below" – do you mean deeper than that density horizon in the firn, or densities less than 550? (likewise "above").

Here, we meant densities less than 550 kg/m3. We have changed the wording to make this clearer. Thank you for the suggestion.

◊ l.162: Please add a statement at the end of the paragraph confirming which values are being used: do you still calculate MO_550 with Eq. 4 and the new parameters in Table 1? or Do you drop the beta and only have a constant term?

We use both the alpha and the beta value and have added a sentence to clarify this.

**2.2.3: Thermal conductivity**

- General comment - I would like more detail about the **latent heat source term** L in equation 5. **How is that implemented in practice? What is the numerical scheme you are using? How do you determine what L is?** E.g., are you using an enthalpy solving method, solving the heat equation for temperature and then making a correction for layers where there is liquid water with temperature below freezing?

We solve the heat equation using the so called "splitting method". In the first half of a time step we solve for water transport using the bucket-scheme. Temperature changes caused by the refreezing of meltwater are added as a source term. This source term equals:

$$\Delta T = \frac{L_f m_{refreeze}}{C_p M_{layer,old}}$$

With $L_f$ the latent heat of melting, $m_{refreeze}$ the amount of water that refreezes inside of the layer, $C_p$ the specific heat capacity and $M_{layer, old}$ the mass of the layer before refreezing. Then in the second half of the timestep no water flux is allowed, making every layer a closed system. We then allow heat conduction to take place by solving the heat conduction equation using the Crank-Nicolson scheme. Due to the nature of the tipping-bucket model, liquid water will immediately refreeze once the layer reaches a temperature below freezing if the pore space is available, otherwise it will percolate towards a deeper layer. We have added a short paragraph to the revised manuscript to clarify this.

- L168: I am confused – you say first on 168 that L includes subsurface absorption of radiation, but then on the next line you say you ignore it.

In the current model setup, G represents the subsurface diffusion of heat through conduction, not the subsurface absorption of shortwave radiation. The text has been reworded to clarify this.

   ◊ l.170 "reasonable approximation" Can you give magnitude of penetration in polar snow and reference supporting this assumption?

Warren et al. (1993) showed that for dry, fine-grained snow most of the penetration takes place in the top few cm of the snowpack. We have also added this info to the text.

   ◊ l.174 "implicit/explicit" use "implicit (respectively explicit) ... in the absence (respectively presence)" instead of "/"

This has been changed according to the reviewer's suggestion.

**2.2.4: Meltwater percolation, retention and percolation**

   ◊ l.196: "as can be seen" add "in Figure 4"

We have changed this following the suggestion.

   ◊ l.202: "Magnusson and others (Magnusson et al. (2015))" Some effort could have been put into the formatting of the citations, especially knowing that the manuscript would go public in the discussion phase.

We apologize for these issues, this has been changed.

   • L202 – citation formatting error

The citation has been changed (see the previous comment).

   • L209: You state that the latter assumption is thought to be valid; what about the first (standing water)?

We meant to indicate that these processes are coupled, i.e., liquid water must first be prevented to percolate downwards and LWC to exceed the irreducible water content before it can run off laterally. We have clarified this by adding the word 'subsequent' between 'standing water' and 'lateral runoff'.

   ◊ l.212-213: Here you mention a sensitivity analysis that is not fully described: how do you evaluate whether it is improving or not the model? Maybe that statement would be more suited in the discussion as an interesting side analysis.

We agree with the reviewer that this paragraph is better suited as a side analysis in the discussion section, so we have moved it there.

**2.2.5: Firn thickness and elevation change**

◊   Section 2.2.5. Here you describe quantities that are derived from the model and not the structure of the model itself. I recommend separating these two in the structure of the method section: one section dedicated to the model and another one dedicated to the model evaluation.

To differentiate between the model description and model derived quantities we changed Section 2.2.5 into Section 2.4 such that Section 2.2 now only discusses the model setup.

•   L219: missing a v for snowdrift – should be vsnd to be consistent with others

The snowdrift is included as $v_{err}$.

◊   l.220 is it normal that v_ice is included in Eq. 8?

$v_{ice}$ represents the rate at which firn turns into ice at the bottom of the column, and as such impacts the thickness of the firn layer. It is assumed equal to the reference period SMB expressed in ice equivalents. We think it is appropriate that we mention $v_{ice}$ in equation 8 in such that we mention all different processes that determine the thickness of the firn layer.

**2.2.6: Model initialization**

◊   l.233 "After the spin-up is finished…." Is this still part of the initialization process? Can you explain why the latest period needs to be run before starting the real run again in 1960?

The wording in the manuscript might have been confusing. The spin-up consists of running the 1960-1980 period multiple times. After this, the real run is started. This is now explained more clearly in the revised manuscript.

•   L245: Can you explain the 3-minute time stepping in more detail? I am not sure what you mean here. Do you run the model at 3-minute resolution and then just save the results every 3 hours?

In previous model version, we used a temporal resolution of 3 minutes when liquid water is present location. Otherwise, we ran the model with a 3 hour time step. Which time step was used, was determined before starting the model run. However, in the new version of the model this has been changed and the model is ran with a 15 minute temporal resolution. The text has been updated.

**3.1 Firn density**

◊   l.251 The statistics of these 29 cores need to be presented separately because they are used in the calibration of the densification scheme.

This will be added as a supplementary material.

- L256: "here the FAC is calculated" Do you mean that Equation 9 calculates the FAC over the entirety of the firn column?

Since each firn core has a different length, we calculate the FAC for the entirety of the available firn core. For example, if a firn core goes 20 m deep, we calculate the FAC over the top 20 m. However, if a different core goes 60 m deep, we calculate the FAC for that location over the top 60 m. As a result, each point in Fig. 5 represents FAC calculated over a different depth. We opt for this instead of using a fixed depth since not all observations go up to the same depth, but we want to compare the model results to as many observations as possible. For us, what is important is that the modelled and observed FAC over the whole range are as close as possible regardless of the depth and not only up to a fixed, arbitrarily chosen depth. We will reword this section to clarify this.

- L258: FAC<15 melt – is this an observation from you model results? Say so.

This statement is indeed based on model results. We will add that to the manuscript.

- L261-265: I am confused – the numbers you write in this paragraph are different than those shown in Figure 5.

The numbers in Figure 5 are the bias and RMSE of all cores in the figure. The numbers in this paragraph are the bias and rmse of cores with FAC < 15 m and FAC > 15 m respectively, to differentiate between high and low melt locations. We will make this distinction clearer in the updated manuscript.

- Figure 5 and L264: define what 'Bias' means specifically, i.e. how are you calculating it, and what is it actually a measure of?

The bias is the mean difference between the modelled and the observed FAC. It quantifies the mean over- or underestimation of the model. We will add an additional sentence explaining this.

◊ Figure 5. Please define R_MA in the caption and in the text. Please sort out the legend items so that "old" comes before "new" (here and elsewhere). Statistics for calibration cores and evaluation cores should be presented separately, either here or in the text.

We have changed the order of old and new as the reviewer suggested. The statistics for the cores will be added as supplementary material.

◊ l.265: I recommend not using "/" unless it means "divided by". Change to " decreased from -0.40 to 0.61 m and from 2.14 to 1.32 m, respectively." It actually shows that the mean bias actually does not decreased in absolute values. Use "change" instead or rephrase.

The suggested changes have been made to the manuscript.

◊ Figure 6: For all panels, please limit the y-axis to the observations. Please present the old before the new (here and everywhere else in the manuscript).

The figures and text have been changed such that the old results are presented before the new results in the figures and in the text. The new range of the y-axis has been changed to be the same as the depth of the observed profile.

• General comment: I was surprised to see the **firn aquifer site** chosen as a test site because the authors state explicitly (line 209) that their model does not simulate standing water on ice layers, which is what is occurring in the firn aquifer zone. W**hy should one expect the model to perform well** at a site where it is not configured to simulate the observed conditions?

This is a fair remark, in the revised text we added the following explanatory sentence: "The aquifer site was selected because its facies represent a distinct climatological zone on the GrIS, with a combination of high melt and high accumulation, which we expect will results in distinct firn characteristics. Standing water is not allowed in IMAU-FDM v1.2G, while this is known to occur at firn aquifer sites (Koenig and others, 2014, doi:10.1002/2013GL058083; Miege and others, 2016, doi:10.1002/2016JF003869; Montgomery and others, 2017, doi:10.3389/feart.2017.00010; Miller and others, 2020, doi:10.1029/2019WR026348), so that modelled LWC remains a qualitative rather than quantitative estimate. In spite of this, it has previously been shown that the model accurately reproduces the spatial distribution of aquifers (Forster and others, 2014, doi:10.1038/ngeo2043), from which we conclude that first order processes that determine temperature and pore space (FAC) are sufficiently represented."

• L261-265: You should be more consistent delineating 'FAC in dry' vs 'low FAC' either use wet/dry or high/low, and state your thresholds

In order to avoid confusion, we have updated the text to only refer to 'low' and 'high' FAC instead of wet/dry.

• L263: why did it switch from underestimation to overestimation?

This can be attributed to the new fresh snow parameterization, which results in lower densities, especially close to the surface. We have added this explanation to the text.

• L267: here is another example of confusion with old/new – in this case you are just referring to the old/new densification equations?

Here, we are referring to the old and new model version. We have changed this to v1.1G and v1.2G respectively to avoid confusion.

• l.270: Consider naming this "core-specific RMSE in firn density"

This is a good suggestion, and we change the name in the text to avoid confusion.

- Eq. 10: Isn't this the RMSE? Why are you referring to it as cost function and calling it Phi, and then pivoting in line 271 and calling it RMSE (and then again using Phi at line 273)?

This parameter is different from the RMSE discussed earlier. In the text, RMSE refers to the error in the FAC:

$$RMSE = \sqrt{\frac{1}{n}\sum (FAC_{model} - FAC_{observation})^2}$$

Where n is the number of observations. The cost function ($\Phi$), however represents the RMSE of a single profile:

$$\Phi = \sqrt{\frac{1}{k}\sum (\rho_{model} - \rho_{observation})^2}$$

Where k is the number of layers in the observed profile. The mean $\Phi$ is then used as a metric to evaluate the model's performance:

$$\Phi_{mean} = \frac{1}{n}\sum \Phi$$

As mentioned in the previous comment, the naming will be changed and the text will be reworded in order to make this clearer.

◊ l.274 "improved" please provide the phi for these two examples. The improvement is not obvious.

We will add this to the text.

- L271-276: Is the improvement of v1.2G over the "old" model, as demonstrated in Figures 5 and 6, due to the new densification equation, or due to the improvement of the surface density parameterization?

Since surface density, densification rate and conductivity all influence each other, it would require extensive sensitivity tests of all different permutations of the parameterizations used to quantify how each change in the model formulation contributes to the improvement. We did not perform these, since the focus of the paper is on the improved performance by the combination of these three changes, and not on investigating the model's sensitivity to each individual change. Therefore, it is unfortunately impossible to give a quantitative answer to this question. However, we know that changes to the new snow density parameterization are most important near the surface and leads to lower densities there. Simultaneously, the $MO_{550}$ values are larger and result in faster densification and partly offsets the smaller densities near the surface. The combined effect of the two results in overall larger FAC, as shown in Fig. 5, which agrees better with observations. Since a lower surface density leads to a lower FAC and a higher densification rate leads to a lower FAC, this suggests that the lower surface density leads to a higher FAC, whereas the new densification rate ensures that the firn profile is modelled correctly at greater depths.

◊ l.282 "main reason…" I am not sure how to interpret this sentence. What is the instantaneous surface density? Please provide statistics (phi_old, phi for a constant 315 kg m-3 density and phi_new for both DAS2 and FA-13) to support this type of statement.

With 'instantaneous', we mean that a surface density parametrization that depends on the air temperature at the instant the accumulation event. In the updated model we instead use mean annual temperature. The text has been reworded to make this clearer, and the words "main reason" will be removed. The phi belonging to the locations in the figures will be added to the text.

◊ l.288: "cold bias…" Please split this sentence in two and please give the mean bias values to justify that the cold bias has been reduced.

As the reviewer suggested, this sentence will be split into two and the mean bias will be added to the text.

**3.2: Firn temperature**

◊ l.290 "The main reason…" How do you identify the main reason? Did you make a sensitivity experiment (like old model + conductivity from Calonne et al. compared to old model + updated densification scheme or compared to old model + new fresh snow density parameterization)?

Please see our response to the comment by Stevens on L.271-276. We will rewrite this sentence and avoid the words "main reason".

◊ l.294: "cold bias in RACMO2" Can you give more details about this bias? has it been described in other studies? Also considering the next sentence: is there the same structure in the bias of the surface forcing (cold bias for temperate and warm location and warm bias for the very cold location)? I remember that the ablation area reaches rather high in RACMO in western Greenland (Steger et al. 2017) could it be that some of the observation sites where refreezing and latent heat release warm up the firn when in the model it is actually pure ice and does not see refreezing? Then the cause of the bias is not a cold bias at the surface.

Thanks for pointing this out. This statement in the paper turned out to be erroneous: it referred to the cold bias in the previous RACMO version (RACMO2.3p1) which has been resolved in the current version RACMO2.3p2 (see e.g., Van Wessem and others, 2018, doi:10.5194/tc-12-1479-2018; Noel and others, 2018, doi:10.5194/tc-12-811-2018). The reviewer suggests that RACMO2 has a too extensive ablation zone in west Greenland, leading to a lack of firn to refreeze in, but this also was resolved in RACMO2.3p2 (Ligtenberg and others, 2017, doi:10.5194/tc-12-1643-2018). We therefore are unsure as to the exact reasons for the remaining temperature biases, which likely is a combination of uncertainties in the forcing and uncertainties in IMAU-FDM. This is now expressed more clearly in the revised text: "In spite of the clear improvement, a cold-bias remains for some of these

locations, while for the low-melt locations (T10 < −20 ◦C), a persistent warm model bias remains. The remaining temperature bias can come from uncertainty in the forcing (surface temperature, liquid water input, snow accumulation, surface density) and uncertainties in the firn model (heat conduction, meltwater percolation, pore space availability, depth of refreezing). Further research is needed to clarify the exact reasons for these remaining biases."

⋄ Please mention that the model does not include firn ventilation, which can warm or cool the firn depending on the season (Albert and Shultz, 2002).

Steger, C. R., Reijmer, C. H., van den Broeke, M. R., Wever, N., Forster, R. R., Koenig, L. S., Munneke, P. K., Lehning, M., Lhermitte, S., Ligtenberg, S. R. M., Mi ge, C., and No l, B. P. Y.: Firn meltwater retention on the Greenland ice sheet: A model comparison, Front. Earth Sci., 5, 3, https://doi.org/10.3389/feart.2017.00003, 2017.

Albert, M.R., & Shultz, E.F. (2002). Snow and firn properties and air-snow transport processes at Summit, Greenland. Atmospheric Environment, 36, 2789-2797.

Thank you for the suggestion. We have added this remark as well as the citation to the text.

- L293: I would expect that a cold bias would be more due to incorrect handling of meltwater (e.g. not enough refreezing, so not warm enough) rather than due to a cold bias in the RACMO forcing, especially given that cold sites have a warm bias. Can you comment on this? Regarding the warm bias, does this mean RACMO is biased warm in the colder areas, despite you saying that RACMO is biased cold in the previous sentence?

Please see our answer to Reviewer Vandecrux (l.294) above.

- L295/Figure 8: Indeed, the new model does slightly better than the old, but it appears to be an incremental improvement, and the bigger issue is that there is a still a substantial misfit compared to the observations (more than 5 degrees at 2 m), especially at Summit in the summer. Please provide discussion on this misfit, and the implications it has on your other results (i.e., a temperature difference of 5 degrees will substantially alter the densification rate predicted by your densification equations).

Please see our answer to the comment by Vandecrux (l.294).

- L301: Why do you not include shallower observations of temperature, which are available for DYE-2? Wouldn't this provide an additional metric to test how well the model is capturing the latent heat release due to refreezing?

While comparing to shallower measurements would indeed give more insight into the latent heat release, the dataset from Harper et al., 2012 is unique because it measured the temperatures also at greater depths. We are interested in the temperature not just in the

top 5 metres of firn, but also in modelling the deep firn, adequately representing the deep firn temperature is very important. For these reasons, we have opted to use this dataset.

- L301: How much of the difference in modeled temperature is due to the (a) new conductivity parameterization, (b) the new surface density, and (c) the new densification equation?

We refer to our response to the comment by Stevens on L.271-276

◊ l.304: Since you mention earlier that a potential bias of the surface forcing may be the cause of the bias in the subsurface, can you compare the air temperature in RACMO2.3p2 and as measured by the AWS at these two sites. This will illustrate the potential surface bias unambiguously.

Unfortunately, especially under very stable conditions, T2m as measured by AWS is not a valid measure for surface temperature, as a bias in the surface-to-air temperature gradient can partly compensate or exacerbate the difference. At Summit, where no significant melt occurs, we can with certainty ascribe the temperature difference at depth to the Ts in the forcing, and hence to errors in the surface energy balance (clouds, turbulent fluxes). We added the following sentence in the text: "We deem the differences of 1-2 K at these locations acceptable in light of the potential uncertainties in both forcing and firn processes, as described above."

**3.3: Liquid water content**

◊ l.319: "agrees better" Please justify your statement by numbers.

The mean old and new bias of the volume fraction and penetration depth will be added to the text to support this claim.

- L319: Figure 9 shows that your model is predicting penetration depth that is too shallow by a factor of 3 – but your text focuses on the improvement of the new model over the old. Please add text describing (or hypothesizing) the remaining deficiencies in the model that cause it to fail to predict the penetration depth accurately.

As can be seen in Fig. 8, the temperature at Dye-2 is still too low compared to the measurements despite the improvements. This means that the liquid water will reach firn freezing point at a shallower depth, which most likely is the cause for the difference between the modelled and observed penetration depth. As we have mentioned in our response to the comment by Vandecrux (l.304), remaining errors in the temperature profile come from a combination of uncertainties in the forcing (surface temperature, liquid water input, snow accumulation, surface density) and uncertainties in the firn model (heat conduction, meltwater percolation, pore space availability, depth of refreezing).

**4.1: Summit**

o   General comment: **evaluating the statistical significance (with respect to the interannual variability) of the differences between the new and old version of IMAU-FDM will be useful.** Are the new results (in particular for elevation, temperature and density) significantly different than the former ones? Is it a new major version of IMAU-FDM?

To quantify the difference between the new and the old section, the mean difference between old and new velocity components will be added to the text.

◊   l.321: Remove "pilot application". If an analysis is presented, it needs to be thorough. However it does not need to be long. Indeed, in this section, you dedicate more than 5 pages, 3 plots (15 panels) and one table to an analysis of simulated surface height at three sites. This seems a bit disproportionate. Some easy updates would make it more concise:

We change the section title (see below) which involved removing the word 'pilot'; the idea behind its use had nothing to do with a lack of thoroughness, but rather to indicate that this section presents a first look at the firn depth results rather than an in-depth study of elevation changes in Greenland.

- Right now the climate at each site is described both at the beginning of the section and within each site's subsection. This is redundant.
- The surface temperature panel in each of the three plots are not mentioned in the text. Could be simply removed.
- Accumulation and melt panels could be either moved to the supplementary, or simply removed and replaced by meaningful statistics in the text (e.g. when mentioning variability, you could give the standard deviation, when mentioning a low melt or accumulation period, you could give the average for that period, which can be compared to the long term average values in Table 2).
- With the two previous updates, Figure 10, 11, 12 can be merged.
- In the text, a lot of the description is redundant in the three sites: fresh snow has the same impact on all three sites and melt the same impact at KAN_U and FA13. This leads to many redundant sentences (l. 374 "Just like Summit..." l.386 "a similar picture emerges..."). Analyzing the three sites in one section and using a process-oriented structure (rather than site-oriented) would cut down a lot of text.

Thank you for these constructive comments. Below we explain how we plan to address these points in the revised manuscript:
- Reviewer Stevens was interested in specific details of the time series in relation to firn thickness, and this was also our intention of this section. To make the aim of this section clearer, we changed its title into: "Connections between surface climate and firn layer thickness."
- The goal of this section is to discuss the temporal variability of firn characteristics as a function of variability in (surface) climate. To explain these properly, one needs to show time series of accumulation and Ts relative to the period mean (dashed lines). Moving these

time series to the supplementary material would force the reader to flip back and forth, which we would like to avoid.
- In contrast to what the reviewer states, (surface, firn) temperature is mentioned in the text, to explain e.g., variations in the dry firn compaction rate. We will make the reference to the figures more explicit.
- We prefer to keep the figures as are, i.e., not merge them, which we feel would make them cluttered. We will, however, redesign them such that they will nicely fit in a column.
- Following the reviewer's suggestions, we will resolve any textual redundancies to make the text more concise, we aim for a 30% reduction in length. We will take extra care not to repeat previously discussed processes that are similar at the three sites.

- o General comment: where the old and new version of the model are compared at 3 sites, its should be interesting to **evaluate the impacts of the new developments in deep. For example, by showing time series of temperature and density at 10m.**

While we agree that such an analysis would be very interesting, this section is already long, as mentioned by reviewer Vandecrux, and adding these plots would make it too long. Therefore, we decide to focus only on the elevation change and its underlying components in this section. Moreover, we believe that a more detailed discussion of these trends is better suited for a different paper and journal.

- • L335: Did you fit trendline to the 1970 to 2000 time series to find the m/yr increase, or are you just differencing the 2000 and 1970 values, and then dividing by the number of years?

Here, we take the difference between 1970 and 2000 and divide by the number of years. This will be made clearer in the revised text.

- • L338: "Nevertheless, the individual velocity components being very different" – sentence structure issue – "are very different"?

This sentence was not worded properly and following the reviewer has been changed to "Nevertheless, the individual velocity components are very different"

- • Figure 10: Your model predicts that the elevation has lowered in the last decade, but surface elevation measurements from Summit (Hawley et al., 2020; sorry to self-reference but it is the dataset I am most familiar with) show that the elevation at Summit increased at 0.019 m/year from 2008 to 2018. Can you explain your model's inconsistency with the observations?

  References:
  Hawley, R. L., Neumann, T. A., Stevens, C. M., Brunt, K. M., & Sutterly, T. C. (2020). Greenland Ice Sheet elevation change: Direct observation of process and attribution at summit. Geophysical Research Letters, 47, e2020GL088864. https://doi.org/10.1029/2020GL088864

Apologies for missing this obvious reference and opportunity for evaluation. Indeed, the difference is very interesting and worth investigating further. We believe that it is mainly caused by uncertainties in the spin-up period. We spin-up the model with the period 1960-1980. This means that the firn layer is in equilibrium with that climate, and its surface elevation change over that period nets to zero. The assumption is that the period 1960-1980 is representative of the past climate. However, this is likely not the case and errors made in the spin-up will result in a drift in de elevation change. As we mentioned in our response to the reviewer's general comment (the second bullet on the first page of this document), we will investigate this sensitivity further along the lines of Kuipers Munneke et al, 2015.

- Figure 10: I am curious about the sudden decrease in elevation in 2019-2020, which appears to be related to the low accumulation in 2019. Is that decrease consistent with altimetry data?

Indeed, both laser and radar altimetry confirm significant surface lowering in this period; these data are currently being prepared for publication by others.

- L340: Given that vsnow and vfc offset each other, is your new model effectively the same from an altimetry standpoint as the old model? By this I mean: is the new formulation a better mathematical representation of physical processes occurring in nature? Or, does 'new' differ from 'old' just in that you added several cores to the calibration and changed the surface density, so the calibration coefficients are different? Does this mean that in using your densification equation, the surface density must be prescribed as you do with Equation 2? If so, can your equation be used to model firn in a location where Equation 2 is not valid?

While it is indeed true that v_snow and v_fc offset each other most of the time, this is not always the case, as can be most clearly seen in locations with a stronger seasonal cycle. With the changes presented in this paper, we try to improve the model both from a 'physical' as well as from an 'altimetry' point of view. Lastly, errors in Eq. 2 will of course propagate into the model results, however, since Fausto et al. (2018) derived their result from a large dataset spanning most of the Greenland ice sheet, we expect it to be robust and widely applicable. This is also reflected in our improved modelling of the FAC.

- l.344 "leads to lower surface density" Can you give an average value for the surface snow density used by the old and new model at that site?

These values will be added to Table 2.

- l.393: please quantify "very different"

The difference will be explicitly quantified in the updated manuscript.

- l.394: Why is the magnitude of the melt different in the new model? That should be presented.

The amount of mm melt taking place is the exact same between both model versions. However, since the new model formulation leads to lower densities near the surface, the same amount of melt will result in a larger change in elevation change. We will add this explanation in the text.

◊ l.394-395: Here this 2.5m difference is a very important update for the use of the IMAUFDM output in altimetry studies. If the new model is closer to reality, the old model missed this 2.5 m lowering due to snow compaction. Altimetry studies using the old model would then attribute this 2.5 m elevation change to a change in ice thickness and to 2.5 m of ice leaving that grid cell. It brings, for the first time an idea of the uncertainty that applies on the firn height change correction product provided by RACMO. It also raises the question: is there other sites where the new model leads to a different trend in 1990-2020 surface elevation? It would be highly valueable to produce a map of difference in 1990-2020 surface elevation trend. This additional analysis would further build trust in the RACMO product for use in altimetry. Making this section more concise would also leave more room for this spatial analysis of surface height evolution.

We agree that this is a very interesting result and exactly the reason why we would prefer to keep the forcing time series in the paper, as well as the aquifer site where this difference occurs, see our previous answers to comments by this reviewer. At the same time, this paper is not intended to discuss at length Greenland ice sheet wide elevation changes; this would require a more detailed comparison with observed volume changes from satellite altimetry and the relative roles of ice dynamics and firn processes. This we consider to be outside the scope of this paper and journal.

o General comment: **adding 2D map** showing the differences of elevation, integrated snow temperature/density (or at 10m) at the end of the simulation (2020) will be useful because I'm note sure if the 3 selected locations are representative of the whole ice sheet.

Please see our response to the comment from Vandecrux l.394-395.

5: Summary

◊ l.408: "predicts predicts"

The repetition has been removed.

◊ l.416: Thanks for sharing the code!

You are welcome! Please note recent updates where some bugs were removed.

◊ l.418: Please add a data availability section with links to freely available data used in the study.

This will be added to the manuscript.

◊ Give a link to and cite the SUMup dataset:

Lora Koenig and Lynn Montgomery. 2017. Surface Mass Balance and Snow Depth on Sea Ice Working Group (SUMup) snow density, accumulation on land ice, and snow depth on sea ice datasets. Arctic Data Center. doi:10.18739/A2W950P44.

Give a link to and cite the firn temperature data:

Baptiste Vandecrux. 2020. Firn temperatures and measurement depths at nine Greenland Climate Network (GC-Net) weather stations, 1998-2017. Arctic Data Center. doi:10.18739/A2833N00P.

Give a link and cite the upGPR data if it has been made available.

These links have been added to the new data availability section.

◊ l.426: Please fix link

Thank you for spotting this mistake; it has been fixed.

◊ l.458: Double entry

The double entry has been removed from the references.

◊ l.479: A more appropriate description of the SUMup data is:

Montgomery, L., Koenig, L., and Alexander, P.: The SUMup dataset: compiled measurements of surface mass balance components over ice sheets and sea ice with analysis over Greenland, Earth Syst. Sci. Data, 10, 1959–1985, https://doi.org/10.5194/essd-10-1959-2018, 2018.

Thank you for pointing this out. We have changed the reference accordingly.

◊ l.578: Correct reference:

Vandecrux, B., Fausto, R. S., van As, D., Colgan, W., Langen, P. L., Haubner, K., Ingeman-Nielsen, T., Heilig, A., Stevens, C. M., MacFerrin, M., Niwano, M., Steffen, K., and Box, J.E.: Firn cold content evolution at nine sites on the Greenland ice sheet between 1998 and 2017, J. Glaciol., 66, 591–602, https://doi.org/10.1017/jog.2020.30, 2020a.

Thank you. This reference has been changed into the suggested reference.

REFERENCES:

Brandt, R., & Warren, S. (1993). Solar-heating rates and temperature profiles in Antarctic snow and ice. *Journal of Glaciology, 39*(131), 99-110. doi:10.3189/S0022143000015756

Brun, E., David, P., Sudul, M., & Brunot, G. (1992). A numerical model to simulate snow-cover stratigraphy for operational avalanche forecasting. *Journal of Glaciology, 38*(128), 13-22. doi:10.3189/S0022143000009552

Fausto, R. S., Box, J. E., Vandecrux, B., van As, D., Steffen, K., Macferrin, M. J., Machguth, H., Colgan, W., Koenig, L. S., McGrath, D.,
Charalampidis, C., and Braithwaite, R. J.: A snow density dataset for improving surface boundary conditions in Greenland ice sheet firn
modeling, Frontiers in Earth Science, 6, https://doi.org/10.3389/feart.2018.00051, 2018.

Harper, J., Humphrey, N., Pfeffer, W. *et al.* Greenland ice-sheet contribution to sea-level rise buffered by meltwater storage in firn. *Nature* **491,** 240–243 (2012). https://doi.org/10.1038/nature11566

Kuipers Munneke, P., Ligtenberg, S. R. M., Noël, B. P. Y., Howat, I. M., Box, J. E., Mosley-Thompson, E., McConnell, J. R., Steffen, K., Harper, J. T., Das, S. B., and van den Broeke, M. R.: Elevation change of the Greenland Ice Sheet due to surface mass balance and firn processes, 1960–2014, The Cryosphere, 9, 2009–2025, https://doi.org/10.5194/tc-9-2009-2015, 2015.

Ligtenberg, S. R. M., Helsen, M. M., and van den Broeke, M. R.: An improved semi-empirical model for the densification of Antarctic firn, The Cryosphere, 5, 809–819, https://doi.org/10.5194/tc-5-809-2011, 2011.

Vionnet, V., Brun, E., Morin, S., Boone, A., Faroux, S., Le Moigne, P., Martin, E., and Willemet, J.-M.: The detailed snowpack scheme Crocus and its implementation in SURFEX v7.2, Geosci. Model Dev., 5, 773–791, https://doi.org/10.5194/gmd-5-773-2012, 2012.

---

## Referee Report (RR1)

Review of (revised version of)
*Improved representation of the contemporary Greenland ice sheet firn layer by IMAU-FDM v1.2G*
by Brils et al.

Reviewer: C. Max Stevens

General comments:

I appreciate the work the authors have done to address the comments from the other reviewers and me on the previous version of the manuscript, and I think the current version is a significant improvement. I recommend that the paper be published after minor edits. I have provided line-by-line comments only. The gist of most of my comments has to do with writing clarity: in numerous places I found explanations to be unclear, and in some cases I was looking for more details about methods or assumptions.

21: the word 'by' occurs 3 times in this sentence

27: I think you mean changing the mass, not changing the mass balance – mass balance in my mind refers to the mass sum of accumulation and ablation processes.

34: Not sure if it is worth mentioning here or following paragraph, but recent work by Rennermalm et al. (2022) suggests that pore-space loss (in SW Greenland, at least) is not entirely irreversible.

50: be specific of what kind of observations – density? Temperature? Depth-age?

52: can you give an example or two of a coupled RCM/firn model?

59, Section 2.1, and Section 3: You define v.1.2G on line 59, but you do not in the paper explicitly define what v1.1G is – is that model version described in Kuipers Munneke et al. (2015)? Given that section 3 is mostly comparing the outputs from the two versions, I think that more detail is needed at the start of section 3 describing the comparison. For example, at the end of the paper I was still not sure if the v1.1G and v1.2G results were both produced using the same version of RACMO (i.e. the same forcing) with different FDM physics, or if the v1.1G is uses a different, older version of the RACMO forcing. A short paragraph describing the two model runs that are being compared will help.

71-72: this neglects to mention Section 5. If you are outlining the paper with this amount of detail I think it is worth mentioning that section as well.

159: This is written as if you have already introduced the new set of observations; I suggest "In order to calibrate Eq. 3 to a new, expanded set of observations (Section X.X), we …". Also, consider rewording here to indicate that MO is changed from the previous model version but retains the same general form.

169: I think there is an issue of confusing wording – I would think that the previous calibration used 22 cores and the new calibration used 29 – but this implies that previous calibration used an expanded data set? Reading forward to section 2.4, it is a bit unclear also – there are 123 observations. Those are used just for evaluation? I think it would be helpful if you added 1-2 sentences in section 2.1.2 describing the new observations used in the present work, or add a bit more detail in section 2.4 (in which case make a reference to 2.4 in section 2.1.2) about the data that were used for the new calibration. (I am not suggesting you list all the calibration cores; rather, just add a bit of text differentiating the calibration data and the evaluation data. E.g., are the calibration cores a subset of the evaluation data, or do you keep them separate?)

176: I think this is what you get at in the following paragraph, but you could be more explicit here: the very low $r^2$ value for $MO_{550}$ indicates that the linear model is not any better than just using the mean of the data (0.67). So: does the new model use the $\alpha_{new}$ and $\beta_{new}$ for $MO_{550}$, or do you just use a value of 0.67? If the former, can you further justify your choice given the low $r^2$?

239: I agree with you that this is probably a fine assumption to make on the spatial scales you are looking at – but I would appreciate a bit of discussion on the implications of that vis-à-vis the discussion of ice lens formation in your introduction: can thick ice lenses/slabs form while using this assumption, or is it necessary to be able to include ponded water?

298: Ok, now I see that you are using 92 cores and the fitting cores are included. I think it would clarify your paper if you add a bit more detail in Section 2.4 about the observations and how you are using them. For example, I was expecting that you used 123 cores based on reading the start of section 2.4. Now I see that the number 123 refers to observations in general – it would be much clearer if you specified e.g., 'we used 92 depth-density profiles, X depth-temperature profiles, and one observation of meltwater intrusion', or something along those lines.

306: "up to a depth" is colloquial and somewhat self-contradictory (up is the opposite of depth perhaps?).

310: comma after FAC, and remove word 'obviously'.

Figure 10: is it possible for you to label the study sites on the figure?

331: please add units on 2.0x10^3. (And note: I think these should be formatted with a latex \times rather than a dot.)

345: It seems that you are initially talking about Das 2 (336), and then it seems here you are talking about Summit? Which is it? Looking at the figure, I am guessing that you may have mislabeled Summit as Das 2?  If it is indeed Das 2, please consider changing the figure to be for Summit rather than Das 2 to be consistent with your section 4 (i.e., you provide specific information about Summit, but not Das 2.) Likewise, consider changing Figure 1 to be density at one of your 3 case study sites.

351: The new model does fit the upper density better, but it is still a rather large misfit in the upper firn. I would like to see a bit more discussion of what is causing that misfit, or at least acknowledgement that there is a deficiency in the model at this sort of site – and I don't mean to pick on this model in particular, because it is a deficiency in firn models in general probably.

360: I know this was picked out in the previous reviews, and it is still not entirely clear: first you say that RACMO2 has a cold bias, but then you say it has a warm bias. Are you saying that the RACMO biases vary spatially? I think changing the text here a little bit will clarify this significantly. Perhaps the issue is the word 'model' – RACMO2 is a model (RCM), and the FDM is also a model – so when you say "a persistent warm model bias remains", it is not clear if you mean a persistent bias in RACMO or resulting from FDM physics.

371/Figure 8: It is not clear to me that the temperature maximum is the refreezing depth – can you justify further why this maximum is assumed to be the refreezing depth? Further, the width of the blue summer DYE-2 temperature curve at ~1m depth (the maximum) would indicate to me that diffusion has happened rapidly, not slower as you posit; i.e., a melt event would cause the firn to be at the freezing temperature, causing a large temperature gradient, which will diffuse much faster than a smaller temperature gradient. I am willing to believe your explanation, but in their current form the explanations seem incomplete.

392: How are you calculating RMSE of penetration depth and volume fraction? E.g. is one time step in the model compared with one observation over that time period? What is the temporal resolution of the upGPR data? Are you including all of the zero-water periods in your RMSE calculation?

438: change 'like': "… in exceptional years; for example, 1983 was …" (and change to past tense "was")

459: I appreciate the work done for these uncertainty analyses. Can you provide a bit more information that summarizes these sensitivity runs? It is not clear to me from the text for example how many runs were done, and I am curious if the accumulation, melt, and temperature variations were applied in simultaneously in single run? You say one-by-one, but the ensuing sentences do not make it clear if each sentence describes one of those runs or several runs. Perhaps a table in the supplement might work? E.g.

| Run # | Variation |
|-------|-----------|
| 1 | Increased Density by X |
| 2 | Decreased accumulation by Y% |

Rennermalm, Å, Hock, R., Covi, F., Xiao, J., Corti, G., Kingslake, J., . . . McConnell, J. (2022). Shallow firn cores 1989–2019 in southwest Greenland's percolation zone reveal decreasing density and ice layer thickness after 2012. *Journal of Glaciology, 68*(269), 431-442. doi:10.1017/jog.2021.102

---

## Author Response (AR2)

Review of (revised version of)
*Improved representation of the contemporary Greenland ice sheet firn layer by IMAU-FDM v1.2G* by Brils et al.

Reviewer: C. Max Stevens

General comments:
I appreciate the work the authors have done to address the comments from the other reviewers and me on the previous version of the manuscript, and I think the current version is a significant improvement. I recommend that the paper be published after minor edits. I have provided line-by-line comments only. The gist of most of my comments has to do with writing clarity: in numerous places I found explanations to be unclear, and in some cases I was looking for more details about methods or assumptions.

21: the word 'by' occurs 3 times in this sentence

The sentence has been reworded to avoid too much repetition.

27: I think you mean changing the mass, not changing the mass balance – mass balance in my mind refers to the mass sum of accumulation and ablation processes.

Thank you for pointing this out. We indeed mean mass and not mass balance.

34: Not sure if it is worth mentioning here or following paragraph, but recent work by Rennermalm et al. (2022) suggests that pore-space loss (in SW Greenland, at least) is not entirely irreversible.

This is indeed true. However, while pore space loss is not strictly irreversible, in practice the rate at which pore space is recovered is much lower than at which it is lost, especially in regions of low accumulation. Nevertheless, we agree that "irreversible" is too strong a word and have reworded the sentence accordingly. We have also added a reference to the paper of Rennermalm et al. (2022).

50: be specific of what kind of observations – density? Temperature? Depth-age?

Density, temperature and depth-age relation are all commonly interpolated with firn models. These have been added to the text as examples.

52: can you give an example or two of a coupled RCM/firn model?

Examples of such climate models are RACMO and MAR. This has been added to the manuscript.

59, Section 2.1, and Section 3: You define v.1.2G on line 59, but you do not in the paper explicitly define what v1.1G is – is that model version described in Kuipers Munneke et al. (2015)? Given that section 3 is mostly comparing the outputs from the two versions, I think that more detail is needed at the start of section 3 describing the comparison. For example, at the end of the paper I was still not sure if the v1.1G and v1.2G results were both produced using the same version of RACMO (i.e. the same forcing) with different FDM physics, or if the v1.1G is uses a different, older version of the RACMO forcing. A short paragraph describing the two model runs that are being compared will help.

IMAU-FDM v1.1G is indeed Kuipers Munneke et al. (2015). Both models have been ran at the same resolution and are forced with the same forcing, the only difference between the two runs is the model physics. This is now explicitly mentioned in Section 2.1 and Section 2.2, which discuss IMAU-FDM and the RACMO forcing. This way, no entirely new paragraph is needed.

71-72: this neglects to mention Section 5. If you are outlining the paper with this amount of detail I think it is worth mentioning that section as well.

Thank you for pointing this out. We have added a mention of Section 5 to this paragraph.

159: This is written as if you have already introduced the new set of observations; I suggest "In order to calibrate Eq. 3 to a new, expanded set of observations (Section X.X), we …". Also, consider rewording here to indicate that MO is changed from the previous model version but retains the same general form.

The sentence has been changed according to the reviewers suggestion and we have added a sentence to indicate that the form of the MO stays the same.

169: I think there is an issue of confusing wording – I would think that the previous calibration used 22 cores and the new calibration used 29 – but this implies that previous calibration used an expanded data set? Reading forward to section 2.4, it is a bit unclear also – there are 123 observations. Those are used just for evaluation? I think it would be helpful if you added 1-2 sentences in section 2.1.2 describing the new observations used in the present work, or add a bit more detail in section 2.4 (in which case make a reference to 2.4 in section 2.1.2) about the data that were used for the new calibration. (I am not suggesting you list all the calibration cores; rather, just add a bit of text differentiating the calibration data and the evaluation data. E.g., are the calibration cores a subset of the evaluation data, or do you keep them separate?)

As suggested by the reviewer, we added some sentences to section 2.4 in which we more clearly explain that we use 123 observations, 92 of which are firn cores (compared to the 62 cores used in Kuipers Munneke et al. (2015)), one measurement year of liquid water GPR measurements and the remaining observations being measurements of the temperature at a depth of 10 metre. We refer to this section in section 2.1.2.

176: I think this is what you get at in the following paragraph, but you could be more explicit here: the very low $r^2$ value for $MO_{550}$ indicates that the linear model is not any better than just using the mean of the data (0.67). So: does the new model use the $\alpha_{new}$ and $\beta_{new}$ for $MO_{550}$, or do you just use a value of 0.67? If the former, can you further justify your choice given the low $r^2$?

We use both the $\alpha_{new}$ and $\beta_{new}$ for $MO_{550}$ instead of using the mean of 0.67. The main reason for this is that in this way we retain the same general formulation for both MO550 and the MO830.

239: I agree with you that this is probably a fine assumption to make on the spatial scales you are looking at – but I would appreciate a bit of discussion on the implications of that vis-à-vis the discussion of ice lens formation in your introduction: can thick ice lenses/slabs form while using this assumption, or is it necessary to be able to include ponded water?

While it is true that our model does not simulate melting water, it does not prevent the formation of ice lenses in Southwest Greenland. This can, for example be seen in this profile from KAN-U in 2020:

[Figure]

It must be noted, however, that the modelled ice slabs consist of multiple thin layers of high density ice, often interlaid with a thin layers with slightly lower density. In reality, a thick ice slab was formed at KAN-U in 2012 (see for example Rennermalm et al. (2021)). The absence of ponding may be what is causing this behaviour. We have added this discussion to the manuscript, and the firn density profile has been added to the supplementary material.

298: Ok, now I see that you are using 92 cores and the fitting cores are included. I think it would clarify your paper if you add a bit more detail in Section 2.4 about the observations and how you are using them. For example, I was expecting that you used 123 cores based on reading the start of section 2.4. Now I see that the number 123 refers to observations in general – it would be much clearer if you specified e.g., 'we used 92 depth-density profiles, X depth-temperature profiles, and one observation of meltwater intrusion', or something along those lines.

See our response to the comment on line 169.

306: "up to a depth" is colloquial and somewhat self-contradictory (up is the opposite of depth perhaps?).

This has been changed to "to a depth"

310: comma after FAC, and remove word 'obviously'

We have taken over this suggestion.

Figure 10: is it possible for you to label the study sites on the figure?

Although it would definitely be possible to add the names of each study site, we feel that it would clutter the image too much. Instead, the name of every site can be found in the supplementary file. We have added a reference to this list to the caption of the figure.

331: please add units on 2.0x10^3. (And note: I think these should be formatted with a latex \times rather than a dot.)

The units have been added to this sentence.

345: It seems that you are initially talking about Das 2 (336), and then it seems here you are talking about Summit? Which is it? Looking at the figure, I am guessing that you may have

mislabeled Summit as Das 2? If it is indeed Das 2, please consider changing the figure to be for Summit rather than Das 2 to be consistent with your section 4 (i.e., you provide specific information about Summit, but not Das 2.) Likewise, consider changing Figure 1 to be density at one of your 3 case study sites.

Thank you for pointing out the confusing wording. This is a mistake in the manuscript: we should refer here to Das 2 and not to Summit. This has now been fixed. We intentionally did not opt to show only results from the three case study sites discussed in Section 3, since that may give of the impression that these three sites are the main focus of our work and of this manuscript. Instead, we want to demonstrate the model's performance over a wide range of sites. We opted for Das 2 and FA-13 to demonstrate the model's performance at two different climates (wet vs dry). However, we do agree that the density profiles can be insightful and may help interpreting our results. Therefore, we have decided to add the density profiles of Summit, KAN-U and Dye-2 to the supplementary material.

351: The new model does fit the upper density better, but it is still a rather large misfit in the upper firn. I would like to see a bit more discussion of what is causing that misfit, or at least acknowledgement that there is a deficiency in the model at this sort of site – and I don't mean to pick on this model in particular, because it is a deficiency in firn models in general probably.

The modelled density in the upper layers of the firn layer fits the observed density profile better. The old model clearly overestimated the density near the surface. The imporved performance can be attributed to the lower surface density and the new MO fit. Despite the improvement, the densification rate in the upper region is still too high. This may be attributed to the lack of a description of microstructural properties on the firn. In the presence of liquid water the rate at which snow grains grow is increased. Firn with larger grains lead to a lower densification rate. This feedback is currently not present in the model. The presence of liquid water may also reduce the densification rate in a different way: it reduces the effective stress felt by the firn layer, which is the driving force for densification. This process is often observed in soils, where it is called consolidation: initially water takes up a change in stress before the soil matrix. To our knowledge, however, the influence of the pore water pressure on the effective stress has not been investigated in the context of firn densification. We have added this discussion to the manuscript.

360: I know this was picked out in the previous reviews, and it is still not entirely clear: first you say that RACMO2 has a cold bias, but then you say it has a warm bias. Are you saying that the RACMO biases vary spatially? I think changing the text here a little bit will clarify this significantly. Perhaps the issue is the word 'model' – RACMO2 is a model (RCM), and the FDM is also a model – so when you say "a persistent warm model bias remains", it is not clear if you mean a persistent bias in RACMO or resulting from FDM physics.

The bias indeed seems to vary spatially: in some regions the 10 m temperatures are too warm and in others they are too cold. In the manuscript we mention first that there seems to be a cold bias at the warmer locations and a warm bias at the cold locations. We admit that the wording was confusing, and we therefore reworded this section. In the dry interior the error is likely due to an error in the RACMO forcing. In wet areas the error is likely due to missing physics in the way the model handles meltwater and thus refreezing. We have made this more clear in our discussion.

371/Figure 8: It is not clear to me that the temperature maximum is the refreezing depth – can you justify further why this maximum is assumed to be the refreezing depth? Further, the width of the blue summer DYE-2 temperature curve at ~1m depth (the maximum) would indicate to me that diffusion has happened rapidly, not slower as you posit; i.e., a melt event would cause the firn to be at the freezing temperature, causing a large temperature gradient, which will diffuse

much faster than a smaller temperature gradient. I am willing to believe your explanation, but in their current form the explanations seem incomplete.

The maximum in the liquid temperature profile is assumed to be the refreezing depth because from the liquid water measurements we see that liquid water usually penetrates and refreezes around these depths (see Figure 9). Moreover, Summit does not show a similar peak in the temperature profile at that time, indicating that it is likely not caused by the temperature at the surface. Furthermore, the new formulation yields of the conductivity is lower than the old formulation at densities below ~500 kg/m3 (see Figure 3). This, together with the lower snow density, leads to slower diffusion rates. Indeed, the temperature gradients have become larger in both the Summit profile as well as in the Dye-2 profile in the upper 1$^{st}$ metre.

392: How are you calculating RMSE of penetration depth and volume fraction? E.g. is one time step in the model compared with one observation over that time period? What is the temporal resolution of the upGPR data? Are you including all of the zero-water periods in your RMSE calculation?

The zero-water periods are included in the calculation of the RMSE. Since the observations have a higher temporal resolution than the model output, we compare each model time step with the nearest observation. We chose for this approach because we would like to compare the liquid water in the firn column over the whole measurement period. If there is no water detected by the GPR, then we would also like the model to simulate no water in the firn column and vice versa. Therefore, computing the RMSE over the whole period gives the most complete view of the model's performance. We have added this information to the text.

438: change 'like': "… in exceptional years; for example, 1983 was …" (and change to past tense "was")

This has been changed.

459: I appreciate the work done for these uncertainty analyses. Can you provide a bit more information that summarizes these sensitivity runs? It is not clear to me from the text for example how many runs were done, and I am curious if the accumulation, melt, and temperature variations were applied in simultaneously in single run? You say one-by-one, but the ensuing sentences do not make it clear if each sentence describes one of those runs or several runs. Perhaps a table in the supplement might work? E.g.

| Run # | Variation |
|---|---|
| 1 | Increased Density by X |
| 2 | Decreased accumulation by Y% |

In order to make it clearer how we conducted the sensitivity analysis we added a table listing the experiments, as suggested by the reviewer. We also slightly altered the text for clarity. During each sensitivity test, only one of the variables is changed at a time. Then, the resulting uncertainties are added together quadratically.

---

## Author Response (AR3)

Dear Philippe,

To check whether the colour scheme of my figures is suitable for people suffering from colour blindness, I have used the colour blindness test available on Coblis. Luckily, this revealed no potential issues. I would like to thank you for your work as an editor.

Best,

Max Brils